Analysis

# Optimizing Xenium In Situ data utility by quality assessment and best-practice analysis workflows

Sergio Marco Salas [1,2] ✉, Louis B. Kuemmerle [2,3,11],
Christoffer Mattsson-Langseth[1,11], Sebastian Tismeyer[4,11],
Christophe Avenel [5,11], Taobo Hu [1,11], Habib Rehman [2,6,11], Marco Grillo [1,11],
Paulo Czarnewski[7,8,11], Saga Helgadottir[1,11], Katarina Tiklova[1], Axel Andersson [5],
Nima Rafati[9], Maria Chatzinikolaou[1], Fabian J. Theis [2], Malte D. Luecken [2,10],
Carolina Wählby [5], Naveed Ishaque [4] & Mats Nilsson [1] ✉

The Xenium In Situ platform is a new spatial transcriptomics product commercialized by 10x Genomics, capable of mapping hundreds of genes in situ at subcellular resolution. Given the multitude of commercially available spatial transcriptomics technologies, recommendations in choice of platform and analysis guidelines are increasingly important. Herein, we explore 25 Xenium datasets generated from multiple tissues and species, comparing scalability, resolution, data quality, capacities and limitations with eight other spatially resolved transcriptomics technologies and commercial platforms. In addition, we benchmark the performance of multiple open-source computational tools, when applied to Xenium datasets, in tasks including preprocessing, cell segmentation, selection of spatially variable features and domain identification. This study serves as an independent analysis of the performance of Xenium, and provides best practices and recommendations for analysis of such datasets.

Imaging-based methods for spatially resolved transcriptomics (SRT) enable targeted and highly multiplexed detection of individual RNA molecules using fluorescence-based microscopy. These methods are subdivided on the basis of their chemistry into in situ hybridization-based (ISH) (for example MERFISH[1] and SeqFISH[2]) and in situ sequencing-based (ISS) (for example in situ sequencing[3] and STARmap[4]). Similar to the rapid adoption of single-cell RNA-sequencing (scRNA-seq), commercial products based on these techniques could accelerate their dissemination. Several companies have recently launched imaging-based SRT products (for example CosMx by Nanostring, Molecular Cartography by Resolved Biosciences and seqFISH by Spatial Genomics). Among those, Xenium, a product from 10x Genomics based on ISS, claims to generate maps of hundreds of genes at a subcellular resolution. Although Xenium datasets have been used by 10x Genomics to demonstrate the potential of the technology[5]

and benchmark it against specific platforms[6,7], a comprehensive independent evaluation of the platform is still needed. In this study, we explore the characteristics, capabilities and limitations of data from the Xenium platform compared with other SRT technologies. In addition, we generate optimized pipelines for the analysis of Xenium data for several computational tasks, highlighting the biological insights that they can provide (Fig. 1a).

## Results

### Xenium datasets offer high-quality tissue population data

To explore the characteristics of Xenium data, we compiled 25 datasets from 10x Genomics and our own Xenium instruments (Methods), derived from 14 experiments. These samples span a variety of types, altogether representing a total of 1.2 billion reads and 6 million cells (Fig. 1b and Supplementary Table 1). The number of genes profiled per sample type

(the 'panel') ranged between 210 and 392 genes. All datasets included the three-dimensional (3D) position ($x$, $y$ and $z$), gene identity and phred-based quality value ($qv$) of every decoded read, with 81% (range, 72–91%) of the reads on average exhibiting high quality ($qv > 20$) (Fig. 1b and Supplementary Table 1). Using Xenium's default segmentation, an average of 186.6 reads per cell was observed throughout the datasets, with 76.8% of reads being assigned to cells, with no obvious differences between fresh frozen (FF) and formalin-fixed paraffin-embedded (FFPE) sections (Fig. 1b). Only 0.21% of the cells had fewer than ten assigned reads and were excluded from further analysis, positioning Xenium as a suitable platform for assessing cell-type frequencies in tissues.

### Xenium enables reproducible identification of populations

To further explore the Xenium datasets' characteristics, we examined seven adjacent full coronal datasets from mouse brains. Xenium's cell-identification algorithm begins with segmentation of DAPI-stained nuclei, followed by an expansion of the segmentation masks. Using the cell-by-gene matrix of segmented nuclei, we identified 50 cell types that could be mapped onto the tissue to create a cell-type map (Fig. 1c,d and Methods). When assigning these cells to anatomical tissue domains (Methods), we observed a consistent distribution of domain-specific cell types[8] (Extended Data Fig. 1g–i). Datasets generated in independent experiments on similar samples using identical probe panels exhibited a strong similarity in gene-specific detection efficiency, dispersion and reads per cell (Extended Data Fig. 1a–f). Cell-type proportions remained consistent across experiments, with notable differences observed only in the less abundant population owing to biological differences between samples (Extended Data Fig. 1e).

### Xenium retains key 3D and subcellular cell information

Xenium datasets provide the 3D position of each read, yet this spatial dimension is often overlooked during cell segmentation. To address this, we used segmentation-free models in which spatially resolved molecular signatures are identified independently of segmentation. The more parsimonious nature of segmentation-free approaches, which analyze signals as they are detected, enables investigation of local signal properties before the spatial assignment into cells. Using one of these approaches, SSAM[9] de novo mode, we identified 44 cell-type-specific clusters (Extended Data Fig. 2a). This approach consistently linked extranuclear reads to specific signatures without requiring cell segmentation. Xenium's 3D coordinates were used to detect potential mixed-source signals from cells overlapping strongly in the $z$-dimension (Fig. 1e and Extended Data Fig. 2b–e), found in 1.8% of the total cells.

Motivated by the insights that segmentation-free analysis can provide, we next focused on exploring its potential to uncover subcellular patterns. Across datasets, we identified some mRNAs enriched in the nucleus, and others in the cytoplasm (Fig. 1f–h). To systematically identify subcellular mRNA clusters, we applied Points2Regions[10], a second segmentation-free model, on mouse brain datasets (Fig. 1i and Extended Data Fig. 3a). We classified these clusters as nuclear, cytoplasmic or extracellular (Extended Data Fig. 3b). As expected, most of these subcellular clusters exhibited associations with specific cell types. However, we also observed subtle yet distinct expression variations between nuclear and cytoplasmic clusters linked to the same cell population (Fig. 1i and Extended Data Fig. 3c,d), indicating that Xenium's signal density facilitates the in situ identification of subcellular structures. Multiple studies have illustrated how these differences can be used to understand RNA biology and tissue dynamics[11]. Overall, our analysis underscores the value of interpreting spatial datasets as 3D subcellular maps, rather than reducing them to expression matrices with only two-dimensional (2D) spatial information.

### Xenium detection efficiency matches ISH

To gain insights into the limitations and benefits of Xenium compared with other SRT platforms, we conducted a comparative analysis of various quality metrics. Given the comprehensive understanding of the cellular composition of the mouse brain established through many studies using single-cell RNA-sequencing (scRNA-seq)[8,12,13] and various SRT methods[2–4,14–19], this tissue serves as an ideal benchmark. We set out to evaluate Xenium against available datasets from similar areas of the mouse brain. Our analysis included imaging-based SRT datasets generated using open-source technologies, as well as commercial platforms (Vizgen's MERSCOPE, high-sensitivity ISS (HS-ISS)[20], MERFISH[18], Resolve Biosciences's Molecular Cartography and Nanostring's CosMx). For sequencing-based SRT, we used a publicly available Visium dataset[21]. To facilitate a fair comparison of the datasets, cells were resegmented using a common segmentation algorithm (Cellpose[22]), and reads were reassigned to individual cells (Fig. 2a, Extended Data Fig. 4b,g and Methods). To minimize potential segmentation errors that could affect our specificity and efficiency estimates, we used a conservative nuclei-based segmentation approach, wherein only a limited proportion of the detected reads (<10–30%) were assigned to individual cells across platforms. Furthermore, tissues were anatomically annotated, and cells from common brain regions (isocortex, hippocampus and thalamus) were kept for further comparison (Fig. 2a and Extended Data Fig. 4a). Different technologies profiled varying numbers of genes, resulting in very different numbers of reads per cell, with CosMx yielding the highest number (Fig. 2b). Broadly speaking, the reads detected per cell can be increased by simply profiling more genes. Given the current commercial trend of proposing higher-plex panels, these numbers will most likely increase with time; therefore, the analyzed datasets provide only a current snapshot. We chose instead

**Fig. 1 | Overview of the analysis and Xenium's main characteristics. a**, Overview of the analysis performed on Xenium datasets. **b**, Summary table of the Xenium datasets, detailing dataset characteristics, descriptors and quality metrics. IDC, invasive ductal carcinoma; DCIS, ductal carcinoma in situ; ILC, invasive lobular carcinoma; MS, multiple sclerosis. **c**, Uniform manifold approximation and projection (UMAP) of cells in seven mouse brain sections, colored by cell type. ACA, anterior cerebral artery; ARH, arcuate nucleus of the hypothalamus; BLA, basolateral amygdala; BMA, basomedial amygdala; CA1, cornu ammonis area 1; CA3, cornu ammonis area 3; *Car3*, carbonic anhydrase 3; CEA, central amygdala; Chol, cholinergic; CR, calretinin; CT, cortical transition; CTX, cortex; DG, dentate gyrus; ENT, entorhinal cortex; ET, embryonic time; GABA, gamma-aminobutyric acid; Glut, glutamate; Gpi, globus pallidus pars interna; HPF, hippocampal formation; IT, interneuron; LA, lateral amygdala; LH, lateral hypothalamus; L5, layer 5; L6, layer 6; MEA, medial amygdala; MSN, medium spiny neuron; NDB, nucleus of the diagonal band; NP, nucleus pontis; OPC, oligodendrocyte precursor cell; *Otp*, orthopedia homeobox; PAL, pallidum; PF, Purkinje fiber; PH, posterior hypothalamus; ProS, prosubiculum; PSTN, pre-subthalamic nucleus; PVH, paraventricular hypothalamus; Pvp, paraventricular nucleus, posterior part; RSP, rostral superior parietal; RT, reticular thalamus; Scg, superior cervical ganglion; SI, substantia innominata; *Slc17a6*, solute carrier family 17 member 6; STN, subthalamic nucleus; STR, striatum; STRv, striatum ventral part; Thal, thalamus; VLMC, vascular leptomeningeal cells; ZI, zona incerta. **d**, Spatial map of cell types in **c**; replicate 1 is shown. The green square highlights the region of interest (ROI) in **e**. **e**, Spatial maps illustrating 3D coherence in Xenium datasets, including *xy*, *xz* and *yz* views of the ROI. **f**, Box plot of subcellular distribution for genes enriched in nuclei and cytoplasm in mouse brain (left) and glioblastoma (right) datasets. The box plot represents percentiles (0, 25, 50, 75 and 100), excluding outliers, with the center representing the median. **g,h**, Spatial maps showing transcript locations of specific genes in mouse brain (**g**) and glioblastoma (**h**) datasets. **i**, Map of transcripts in oligodendrocytes, colored by Points2Regions cluster in one of the mouse brain datasets (msbrain2). **j**, Box plot of the distribution of the Points2Regions clusters 0, 37, 46, 80 and 89 in **i** in relation to their distance to the nuclei edge. The box plot represents percentiles (0, 25, 50, 75 and 100), excluding outliers, with the center representing the median. **k**, Differentially expressed genes for each subcellular cluster in **i**.

to focus on quantifying gene-specific characteristics of the different assays. We calculated the detection efficiency for individual genes for each technology by comparing read counts obtained for each gene with a reference region-matched scRNA-seq dataset[23]. Our analysis revealed that Xenium was the most sensitive ISS-based technique: its sensitivity was similar to that of ISH-based technologies such as MERSCOPE and Molecular Cartography (Fig. 2c and Extended Data Fig. 4c,e). Notably, all commercial SRT platforms, unlike their homemade counterparts[24],

demonstrated a very similar detection efficiency, highlighting the convergence of platforms in this aspect. For Xenium, this detection efficiency was found to be between 1.2 and 1.5 times higher than that of scRNA-seq (Chromium v2), depending on the metric and region analyzed (Fig. 2c and Extended Data Fig. 4c,e). To further validate these observations, we independently clustered the cells from each dataset using standardized analysis pipelines, identified shared populations and compared gene expression levels across technologies. This

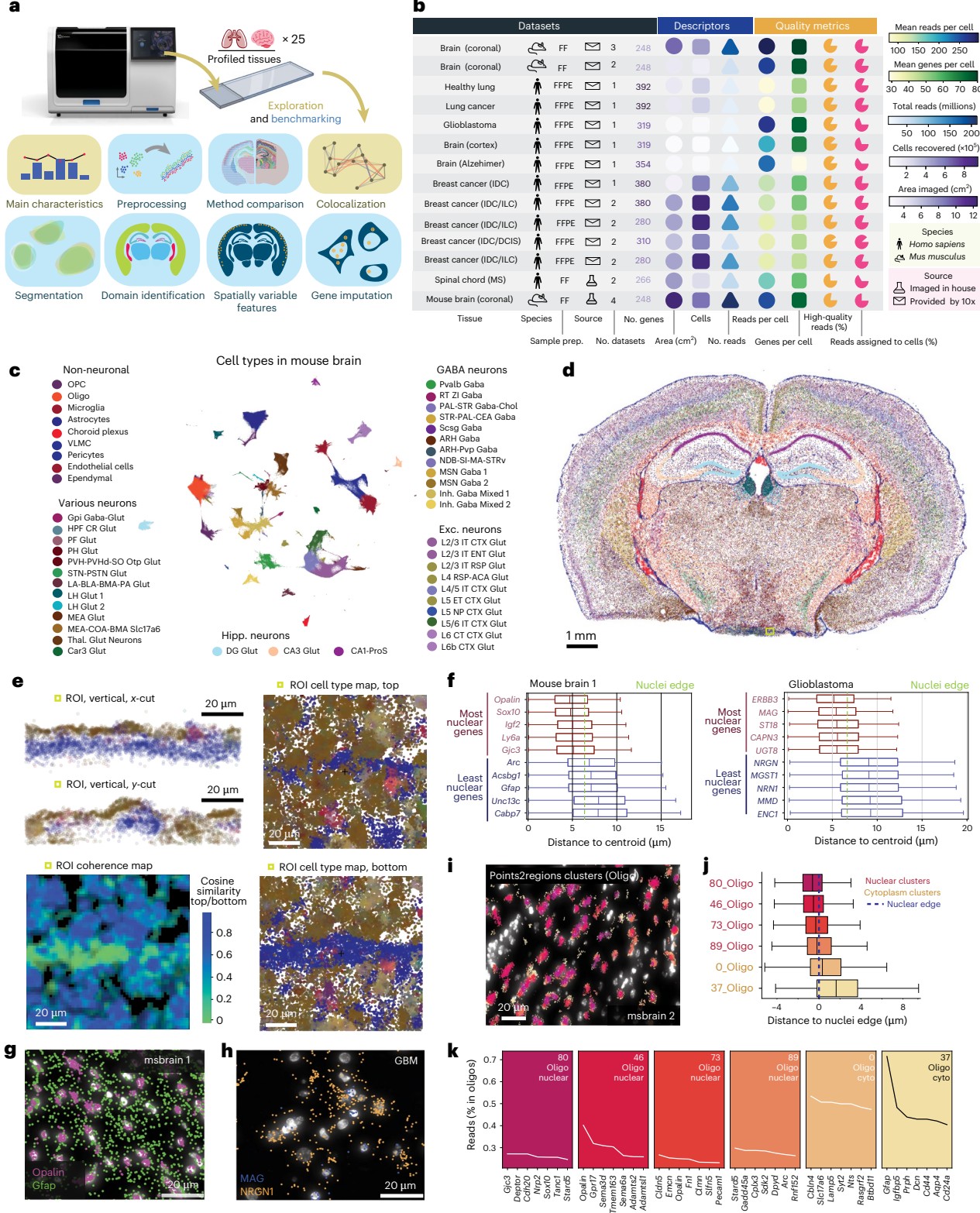

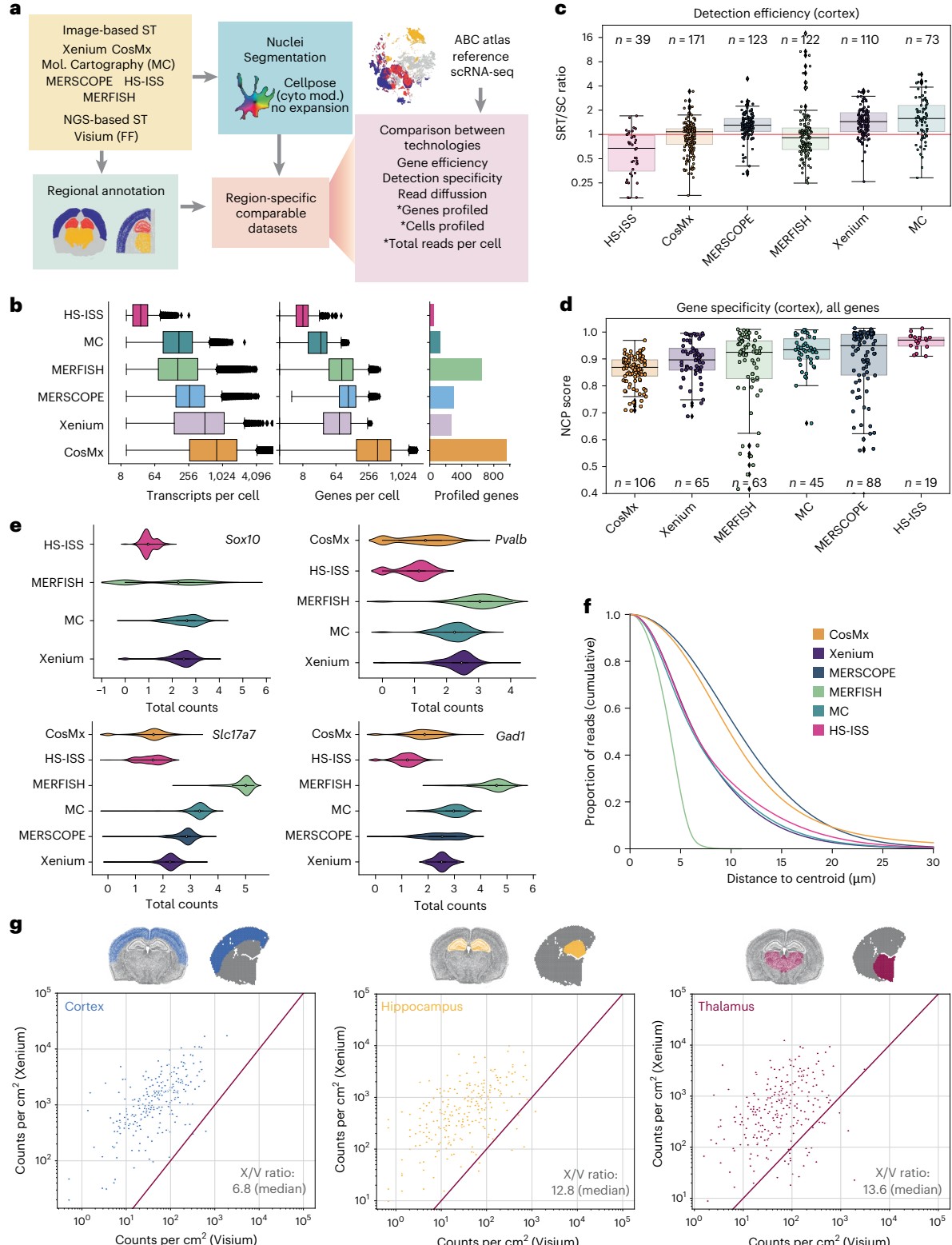

**Fig. 2 | Benchmarking Xenium against other SRT platforms. a**, Overview of the workflow for comparing SRT platforms. In the 'Comparison between technologies' box, the asterisk indicates that the values for the metric differ between experiments. Cyto mod, cellpose cytoplasm model. **b**, Box plot showing the numbers of transcripts per cell and genes per cell for each dataset (left and center). The box plots represent percentiles (0, 25, 50, 75 and 100), excluding outliers, with the center representing the median. The bar plot (right) shows the number of profiled genes per platform. **c**, SRT/scRNA-seq (SRT/SC) gene efficiency ratios for various SRT platforms. Gene efficiency refers to the proportion of transcripts of a certain gene of interest detected using a given platform. The box plot represents gene efficiency in quartiles, excluding outliers, with individual dots showing gene-specific ratios. *n*, number of genes. **d**, Box plot showing NCP scores, ranging from 0 to 1, in quartiles excluding outliers, reflecting the percentage of non-coexpressed pairs in single-cell data that remain non-coexpressed in situ. *n*, number of pairs. **e**, Violin plot of transcripts detected per gene across datasets, focusing on clusters with highest marker expression. **f**, Cumulative proportion of reads by distance from the cell centroid across platforms. Values indicate the proportions of reads that are found at distances greater than or equal to the specified distance. **g**, Scatter plot of reads per gene per square centimeter in Visium versus Xenium across brain regions. X/V ratio, Xenium to Visium ratio.

revealed a comparable number of molecules detected per cell between Xenium and other commercial SRT platforms (Fig. 2e).

To contextualize Xenium's performance alongside sequencing-based methods, we compared it with Visium (Fresh Frozen)[21], the most widely used SRT platform. Given that Visium doesn't have single-cell resolution, we assessed pseudo-bulk gene-specific reads identified by each method for a common anatomical region, normalized by area. Results showed that Xenium was more sensitive than Visium at the tissue level, detecting a median of 12.8 times more reads (Fig. 2g). Notably, some genes that were barely detected by Visium were found to be highly abundant by Xenium.

Efficient detection is crucial in SRT assays, but equally vital is assay specificity. To evaluate the latter, we implemented a metric called negative co-expression purity (NCP), which quantifies the percentage of non-co-expressed genes in our reference single-cell dataset that do not appear to be coexpressed in each SRT dataset. Therefore, a high NCP, close to 1, indicates a high specificity in the dataset. Overall, all the different SRT technologies presented a mean high specificity (NCP > 0.8), with HS-ISS and Molecular Cartography being the most specific technologies (Fig. 2d). The specificity of Xenium was slightly lower than that of other commercial platforms, but was consistently higher than that of CosMx, which presented the lowest values. These results remained consistent after removing all highly expressed genes, which could result in lower NCP scores owing to a broader expression across cells (Extended Data Fig. 4d). Last, we observed differences in the subcellular distribution of profiled reads across different technologies (Fig. 2f). MERFISH and ISS methods displayed a higher concentration of reads near the cellular centroid, whereas ISH-based commercial platforms (CosMx, Molecular Cartography and MERSCOPE) had reads positioned farther away. These discrepancies were evident when comparing individual genes across the various technologies (Extended Data Fig. 4f).

## Nuclear expansion influences cell-type expression profiles

Nuclear expression signatures are typically sufficient for defining cell populations in situ, but incorporating cytoplasmic reads could then enhance cell clustering and labeling. Under this assumption, Xenium's nuclear segmentation is followed by a default radius expansion of 15 μm. This expanded cell-by-gene matrix allows for the identification of cell types that are organized into region-specific clusters, in contrast with the more homogeneous classifications achieved using unexpanded segmentation (Extended Data Fig. 1h). For instance, thalamic oligodendrocytes were grouped together with thalamic astrocytes, rather than with other oligodendrocytes (Extended Data Fig. 1i), indicating that expansion captures domain-specific expression signatures.

To identify the optimal cell expansion, we defined nuclear expression signatures for each cell type and domain-specific background expression signatures (Methods). Our analysis revealed that transcripts located more than 10.71 μm, on average, from the cell centroid exhibited a higher gene expression correlation with domain-specific background signatures than with nuclear cell-type-specific signatures (Fig. 3a,b). This distance likely reflects the average radius of the profiled cells, including both nuclei and cytoplasm. Given that nuclei in this dataset presented a radius of 5.06 μm, on average, the ideal expansion of cells in the samples should be 5.64 μm. However, different cell types presented different optimal expansion distances (Fig. 3b). Thus, segmentation strategies based on the identification of nuclei followed by a rigid expansion might not provide the best solution.

## Baysor and Cellpose outperform standard Xenium segmentation

The influence of cell segmentation on cell-typing accuracy motivated us to explore alternative segmentation methods. We benchmarked the performance of Xenium segmentation against commonly used segmentation strategies (Fig. 3c, Extended Data Fig. 5a,b and Methods).

These strategies can be broadly categorized as staining-based, in which the position of cells is determined by an auxiliary staining such as DAPI (Watershed[25], MESMER[26] and Cellpose[22]); read-based, in which cells are defined on the basis of the read density and composition of tissues (Baysor[27]); or mixed models, in which both staining and the position of reads is used for defining cells (Baysor[27] and Clustermap[28]). Segmentation based on equally distributed bins across the tissue (binning) was included in the comparison as an example of simplistic segmentation. We also applied various cell expansions to each segmentation output (1, 2, 5, 10 and 15 μm).

We next identified groups of strategies that performed similarly (Fig. 3d and Extended Data Fig. 5d). Staining-based strategies using DAPI generated similar outputs, with cell expansion being the force driving their differences. In addition, Baysor-based, Clustermap-based and binning strategies clustered according to method, indicating method-specific segmentation output. We defined the optimal segmentation strategy as the one maximizing the proportion of reads assigned to cells while maintaining specific expression patterns, quantified by negative marker purity (NMP) (Methods) (Fig. 3e and Extended Data Fig. 5c,e). NMP calculates the percentage of detected reads expected in each identified cell type on the basis of a reference scRNA-seq[23]. We found that Baysor-based strategies, particularly Baysor combined with Xenium's nuclei segmentation (BA2P0.8), represent the best segmentation strategy (Fig. 3e and Extended Data Fig. 5e). Moreover, including Xenium's segmentation as a prior results in fewer missed cells (Extended Data Fig. 5b,c). These results were consistent across all datasets (Extended Data Fig. 5d,e).

Finally, we jointly processed the cells segmented using the best strategy (BA2P0.8) alongside those defined by Xenium's nuclear segmentation (Fig. 1c). Although cells defined by Baysor had a higher count per cell, the identified cellular populations were the same across both segmentation strategies (Fig. 3f–h), with mostly mild differences in cell-type abundance. Overall, our analysis indicates that Xenium's default nuclear segmentation masks provide adequate information for defining the main populations detectable in situ, comparable to more-sophisticated segmentation strategies.

## Preparing Xenium data: best practices in preprocessing

To identify populations in situ, cell-by-gene data are typically preprocessed. The preprocessing of in situ datasets involves essential steps, such as filtering out low-quality cells and genes, applying appropriate transformations, reducing dimensionality, and clustering. These steps, derived from the single-cell field, can have a major impact on cell-type identification in situ. Thus, we aimed to define the optimal preprocessing steps for Xenium data.

In the absence of a reliable cell-type reference for Xenium datasets, we used scRNA-seq datasets from Census[29] as our starting point (Methods). Census datasets were transformed to resemble Xenium data by (1) reducing the number of captured genes, (2) varying the detection efficiency of individual genes and (3) introducing the effect of mis-segmentation and technical noise (Fig. 4a, Extended Data Fig. 6a and Methods). Our preprocessing approach involved multiple steps, including scaling, normalization, highly variable feature (HVF) selection and clustering. We considered omitting certain steps and used multiple hyperparameters in each process (Extended Data Fig. 6a and Methods). Finally, we assessed the similarity between the new clusters and the reference labels, identifying preprocessing workflows that maximized accurate cell grouping into clusters. Of note, we found a set of workflows that, when applied, consistently maximizes the similarity between the original and newly generated clusters (Fig. 4b and Extended Data Fig. 6b). The most effective method consisted of: (1) library-size-based normalization, with the total library size set to 100; (2) log-transformation; (3) scaling; (4) the construction of a $k$-nearest neighbors graph using all principal components and 16 neighbors; and (5) Louvain clustering (Fig. 4c). Surprisingly, some top-performing

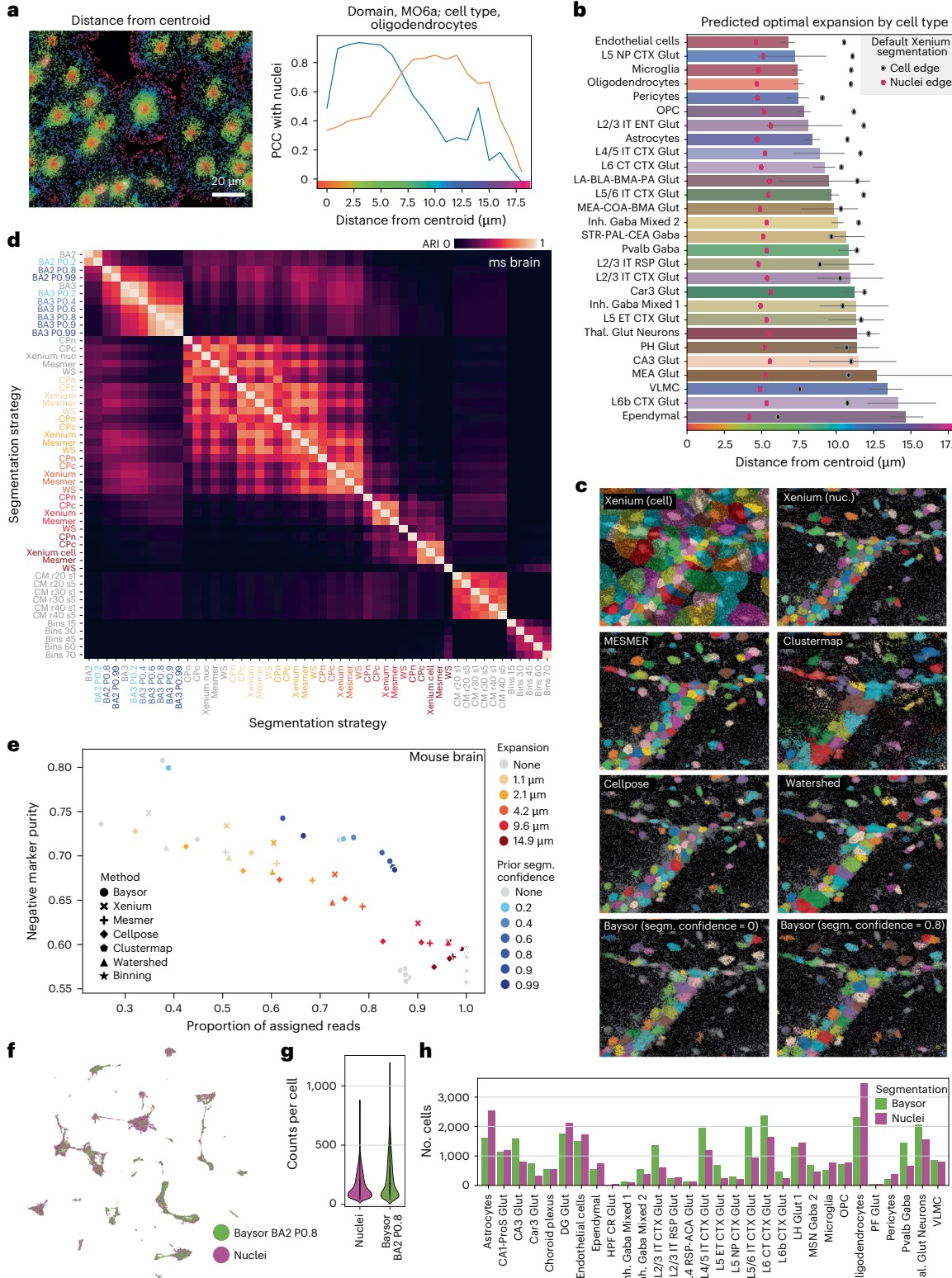

**Fig. 3 | Exploring segmentation in Xenium. a**, Mouse brain region with reads overlaid on DAPI staining, colored by distance to the nearest cell centroid (left). The line plot (right) shows the PCC of oligodendrocytes to the nuclear signature (blue) and to the background signature (orange), depending of the distance to the cell centroid. **b**, Bar plot showing the distance in micrometers of the intersection between nuclei and the domain-specific regions, as in **a**, across cell types. Error bars represent the 95% confidence intervals. Mean nuclei and cell radius are also shown. **c**, Comparison of cells identified with different segmentation algorithms in an ROI (160 × 160 μm), using DAPI background. **d**, Adjusted rand index (ARI) comparison of segmentation outputs (52 top performers) when applied to one of the mouse brain samples profiled (mouse brain section 2). **e**, Scatter plot of reads assigned versus negative marker purity for segmentation strategies applied to mouse brain section 2. Prior segm. confidence refers to the value given to the parameter named 'prior segmentation confidence' in Baysor-based segmentation. **f**, UMAP from coprocessed cells using Baysor and Xenium's nuclear segmentation in a mouse brain ROI. **g**, Violin plot comparing cell counts segmented by Baysor versus Xenium nuclear segmentation methods. **h**, Bar plot of cell counts per population using different segmentation strategies.

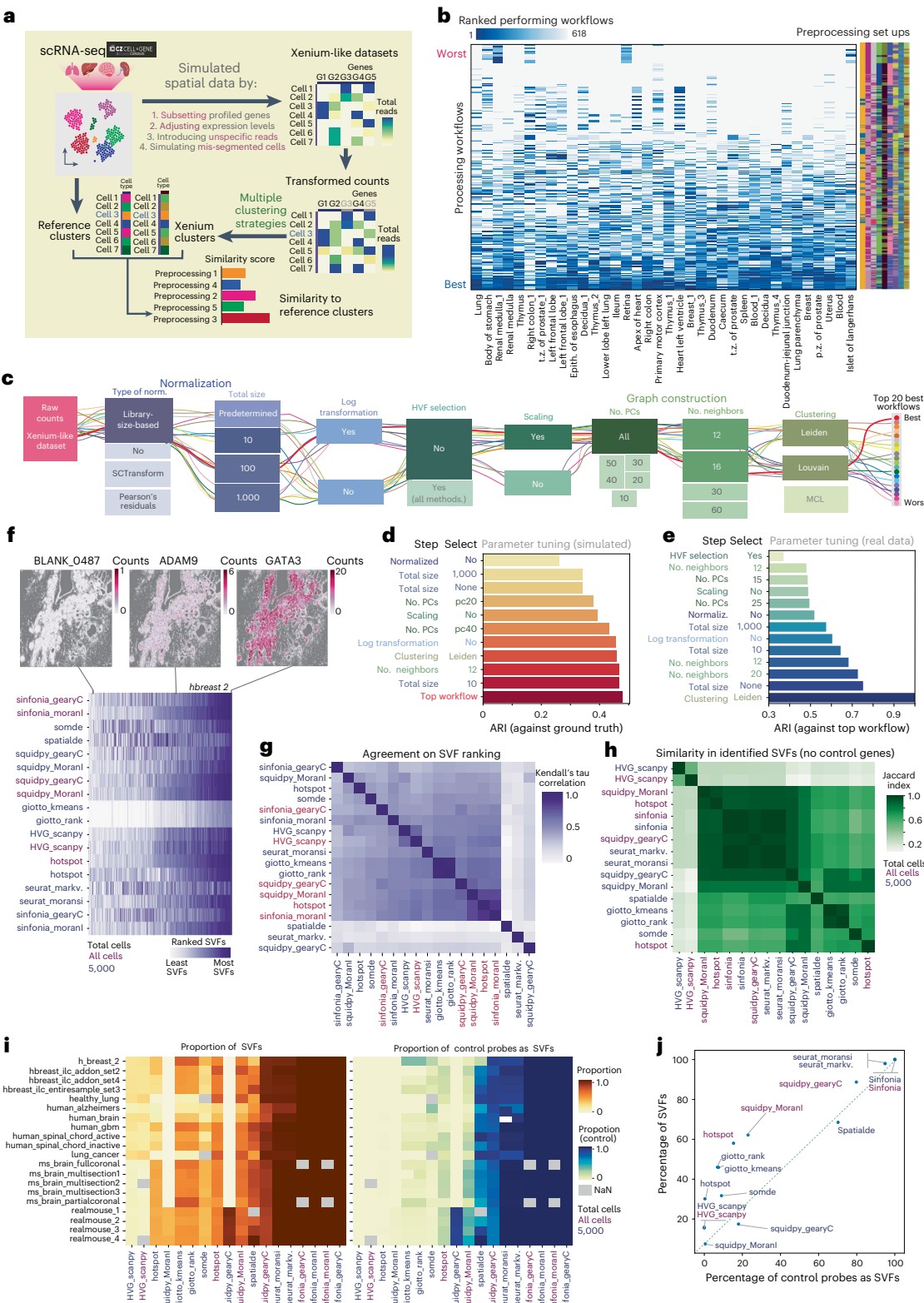

**Fig. 4 | Assessing the best preprocessing methods for Xenium. a**, Workflow
diagram showing simulation of Xenium-like datasets from CELLxGENE Census
single-cell data. **b**, Heatmap ranking preprocessing workflows on the basis of
alignment with reference cell types, with workflows sorted from best (blue) to
worst (white). A summary of the processing setups is included (right), with colors
indicating the preprocessing steps chosen, as indicated in Extended Data Fig 6b.
Epith., epithelium; p.z., peripheral zone; t.z., transition zone; duod-jejunal junct.,
duodenojejunal junction. **c**, Top 20 preprocessing paths, with the best path
marked in red. PCs, principal components; MCV, Markov cluster algorithm. **d**, Bar
plot of ARI, showing the effects of different preprocessing steps on clustering

consistency relative to ground truth. **e**, Bar plot of ARI. comparing workflow
consistency across real Xenium datasets with different preprocessing steps.
**f**, Heatmap of SVF scores across algorithms in a breast-cancer dataset. Example
spatial maps of non-SVF, partial and SVF are shown (top). **g,h**, Mean agreement
(Kendall's tau) (**g**) and Jaccard similarity index (**h**) showing agreement in SVF gene
rankings across datasets. **i**, Proportion of genes (left) and control probes (right)
identified as SVFs across Xenium datasets. Algorithm colors indicate whether
each algorithm used 5,000 cells or the full sample. **j**, Scatter plot comparing the
proportion of control probes with features identified as SVFs by the algorithm,
with colors representing 5,000-cell or full-sample input.

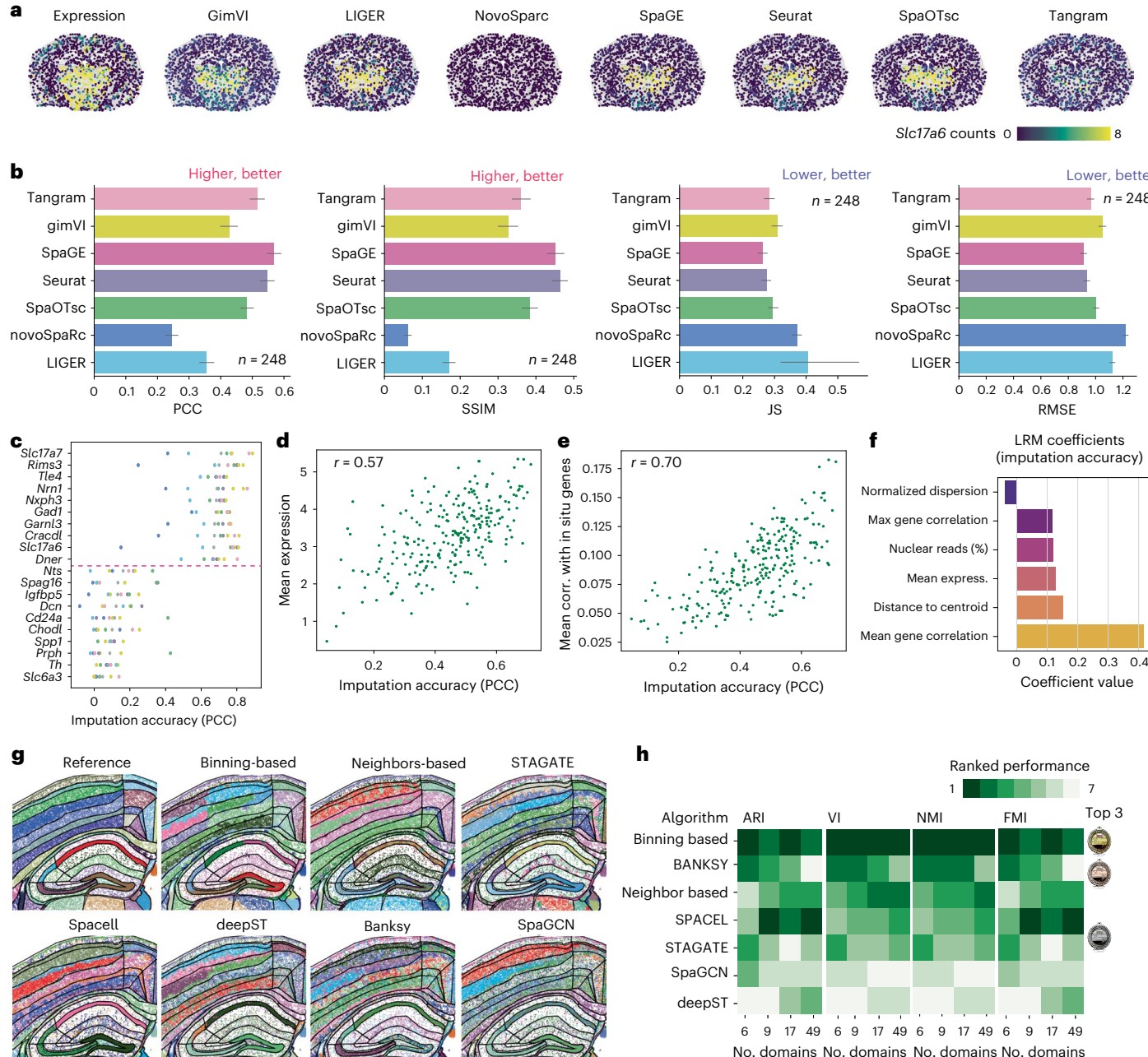

**Fig. 5 | Benchmarking imputation and domain identification algorithms with Xenium. a**, Detected and imputed expression of *Slc17a6* in mouse brain dataset 1 using various imputation algorithms. **b**, Performance of imputation methods assessed with four metrics: PCC, SSIM, JS and RMSE; data are shown as mean ± 95% confidence intervals. **c**, Imputation accuracy (PCC) across genes, highlighting the ten best and ten worst predicted genes. **d**, Scatter plot showing the correlation between PCC for imputation accuracy and mean gene expression. **e**, Relationship between gene-specific imputation accuracy (PCC) and mean gene correlation with other detected genes in situ. **f**, Bar plot of linear regression model (LRM) coefficients indicating feature importance for predicting gene-specific imputation accuracy. **g**, Spatial map of manually annotated domains in a mouse brain section (replicate 1) compared with domains identified by various algorithms. **h**, Ranked performance of algorithms in domain identification using the manual segmentation as a reference, evaluated with ARI, variability index (VI), normalized mutual information (NMI) and Fowlkes–Mallows index (FMI), for different domain numbers[6,9,17,49].

workflows omitted the log-transformation step but maintained scaling, emphasizing that there is no universal best way to process spatial datasets. We next conducted parameter-tuning analysis, taking as a gold standard the previously described best-performing workflow. Using simulated datasets, we identified that the most essential factors in the processing workflow were the normalization method, the library size used in library-size-based normalization, scaling and the number of principal components, selected when building the *k*-nearest neighbors graph (Fig. 4d).

After using simulated data to identify the best preprocessing workflows, we explored the impact on real Xenium data. Using the top preprocessing workflow as a reference, we conducted a parameter-tuning analysis (Fig. 4e). We found that some preprocessing steps, such as HVF selection, normalization and data scaling, were crucial, as their absence led to dramatic differences in clustering outcomes. These results were consistent across datasets, regardless of the metrics used to compute clustering similarity, and closely mirrored results obtained using simulated datasets (Extended Data Fig. 6c–e).

## Selection of spatially variable features using Xenium datasets

As an alternative to HVF selection, identifying spatially variable features (SVFs) is useful in distinguishing genes that explain the main spatial variation patterns within tissue. Given the variety of methods, the choice of algorithm could influence the results. To better understand the differences, we compared the performance of the commonly used methods developed for this task (Squidpy's Moran's I and Geary's C[30], Hotspot[31], SomDE[32], SpatialDE[33], Sinfonia[34], Seurat's mark variogram and Moran's I[35] and Giotto[36]), using full Xenium datasets when possible. Because some algorithms could not be applied to the full datasets owing to their long expected execution times (Extended Data Fig. 7a), we also compared their performance on a subset of 5,000 cells.

Despite the large disparity in the proportion of SVFs identified by each algorithm, we observed a good agreement in gene ranking across most algorithms, with the exception of SpatialDE, Seurat's mark variograms's and Squidpy's Geary's C (5,000 cells) (Fig. 4f,g). However, algorithms differ when classifying genes as SVFs (Fig. 4h–j), with some methods consistently identifying a large proportion of the panel as SVFs (Seurat or Sinfonia), and others selecting a lower proportion (hotspot, Squidpy). Because Xenium datasets include some control probes, which by nature are not going to be spatially variable, we used them to quantify the percentage of false positive features detected by each algorithm. None of the SVF algorithms successfully classified all control probes as non-variable features, with Hotspot being the closest one, selecting <5% of control genes as SVFs. As expected, algorithms that selected a higher proportion of SVFs also reported more false positives, suggesting that the available algorithms might still confuse noisy patterns with spatial variability. Notably, this was not true for the algorithm used to identify HVFs, which detected all control probes as non-variable features while still consistently detecting ~18% of the genes as HVFs (Fig. 4i,j).

## Benchmarking gene imputation tools on Xenium datasets

Targeted SRT methods are typically constrained by the number of genes measured simultaneously. Imputation approaches overcome this limitation by predicting gene expression from a reference scRNA-seq onto a cellular-resolution SRT dataset[37]. We sought to benchmark the performance of seven methods (gimVI[38], SpaGE[39], Tangram[40], Liger[41], Seurat[35], SpaOTsc[42] and NovoSpaRc[43]) using the workflow developed by Li et al.[37] (Fig. 5a and Methods).

Imputation performance was assessed by Pearson correlation coefficient (PCC), structural similarity index (SSIM), root mean square error (RMSE) and the Jensen-Shannon divergence (JS), with a higher PCC and SSIM and a lower RMSE and JS value indicating better prediction accuracy. Using these metrics, we consistently identified SpaGE as the optimal method (Fig. 5b). In addition, Seurat, Tangram and SpaOTsc achieved an overall high performance. Surprisingly, gimVI's performance was lower than one previously reported when using it to integrate scRNA-seq with other SRT technologies[37].

Our workflow compares detected and imputed gene expression of individual genes, making it an effective method for identifying genes with overall low agreement between scRNA-seq and Xenium. By quantifying gene-specific differences using the PCC across all genes and methods, we observed an enormous difference in the imputation performance between genes, consistent across imputation methods (Fig. 5c). We discovered that, among the characteristics we examined, expression level and overall correlation with other genes in the panel were the most strongly associated with effective imputation. By contrast, other factors, such as the subcellular localization of the transcript or its variability, did not significantly affect the imputation performance. (Fig. 5d–f)

## Assessing computational tools to explore tissue architecture

Identifying the architecture of tissue can be helpful to understand its function. Identifying reliable tools for defining these domains is of great interest, yet no independent comparison is available. Therefore, we benchmarked five domain-identification algorithms (Banksy[44], DeepST[45], SpaGCN[46], SPACEL[47] and STAGATE[48]) against the regions identified through expert manual annotation from the coronal P56 section of the Allen Brain Atlas[49] (Fig. 5g and Extended Data Fig. 7b). We also included two simple methods to identify cell compartments (binning-based and neighborhood-based[50]). We found that the domains predicted by binning-based clustering consistently exhibited the highest similarity to manual annotations, outperforming more-sophisticated algorithms (Fig. 5h and Extended Data Fig. 7c). However, these findings might be influenced by the specific architecture of the tissue type analyzed, meaning that the performance of various methods could vary in different tissue types.

## Best practices for processing and analyzing Xenium datasets

On the basis of the evidence presented here, we propose an optimal approach for processing and analyzing Xenium datasets. We have condensed this information in an end-to-end pipeline with the aim of helping Xenium users to maximize the value of their data.

In brief, taking the data obtained from Xenium as an input, first we would re-segment the cells. The optimal algorithm involves two steps: first, identifying nuclei using Cellpose[22] and second, assigning reads to individual cells using Baysor[27]. Cellular expansion is unnecessary, because extra-nuclear reads are assigned directly by Baysor. If cellular segmentation results in poor performance, segmentation-free methods such as SSAM[9] or Points2Regions[10] can be used to identify molecular signatures without identifying individual cells. After segmentation, a cell-by-gene matrix is generated, which can be taken as an input for cell-type identification through standard scRNA-seq workflows consisting of (1) cell filtering, (2) log-transformation and normalization, (3) identification of the main principal components, (4) dimensionality reduction and (5) clustering. To rank SVFs, various algorithms can be used; however, they might yield inconsistent results in their selection of spatially variable genes. In addition, if scRNA-seq or single-nucleus RNA-seq is available, gene imputation can be performed using algorithms such as Seurat, SpaGE[39] or Tangram[40]. Finally, for the identification of domains, using binning-based strategies or algorithms, such as Banksy[44] or SPACEL[47], can be a convenient solution.

## Discussion

In this study, we present an independent exploration and evaluation of Xenium in situ datasets. Xenium can be used to generate highly multiplexed spatial gene expression maps with subcellular resolution. The identification of hundreds of reads per cell, combined with the extensive tissue characterization, facilitates the easy identification of cellular populations in situ.

The detection efficiency of Xenium was comparable to that of Chromium v2 and slightly higher than that of other commercially available platforms, while maintaining a high specificity. These features are consistent across samples, as shown in recent studies[6,7]. Notably, however, most recently launched SRT platforms demonstrate similar performance in most metrics, with the most important difference being the number of profiled genes, a metric expected to change over time as new gene panels are introduced. To enhance our understanding of the strengths and weaknesses of each assay, further independent comparisons of technical aspects, such as imaging time, experimental costs, user-friendliness and reproducibility, are crucial. Several processing steps were identified to be crucial for analyzing Xenium datasets, with segmentation highlighted as one of the most important. The current segmentation provided by 10x Genomics effectively identifies cell nuclei and gathers sufficient information to identify the main cell populations in the analyzed sections. Cell mask expansion after segmentation can negatively impact the characterization of cell populations by misassigning reads to neighboring cells.

We found that Baysor in combination with Cellpose segmentation outperformed the rest of the strategies, effectively defining individual cells on the basis of both the density and identity of individual reads in situ. This strategy enables Baysor to identify cells of varying sizes and compositions; it can even detect cells whose nuclei are not captured in the section. This can happen, for example, when the sectioning plane separates a nucleus from the bulk of its corresponding cell body. Overlapping cells present a common but challenging segmentation issue, owing to the 2D nature of most of the algorithms used. Thus, implementing new segmentation algorithms that account for the 3D structure of the data and incorporating additional staining for cellular membranes would facilitate the correct identification of individual cells. In this context, segmentation-free cell-typing methods represent an alternative to the typical segmentation-then-clustering workflow, enabling identification of cell populations and subcellular patterns. The exploration of subcellular patterns represents an underexplored layer of information that is now becoming accessible thanks to the sensitivity and resolution of new imaging-based SRT methods, such as Xenium.

Given the extensive range of available methods, only a subset of the most popular techniques was included in the benchmarking process for various tasks; therefore, the top-performing method for each task might not be the best one available. In addition, some published algorithms approached the tasks in an alternative manner, resulting in non-comparable outputs (for example, SpaGCN[46] identifies domain-specific SVFs) and were not suitable for benchmarking. Overall, although some efforts have been made, a more systematic benchmarking of these and other algorithms using datasets generated in different tissues with different experimental designs would be beneficial to decipher when to use each algorithm. To achieve this, new and more diverse datasets are essential.

In summary, Xenium represents an overall improvement over other ISS-based technologies. Its increased detection efficiency and high specificity, together with its resolution, enable the identification of cell types in their spatial context, making it a useful tool to explore spatial biology.

## Online content

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

## Methods

### Xenium experiments

Fresh frozen male mouse brain tissue sample (no. 676) from an 8-week-old C57BL/6 strain was purchased from Adlego Biomedical AB/Scantox under ethical permit 16316-2022 and profiled using in situ Xenium platform (10x Genomics). The predesigned 248-gene Xenium Mouse Brain Gene Expression panel was profiled across the 4 sections by the In Situ Sequencing Infrastructure Unit (Science for Life Laboratory), where probe hybridization, ligation and rolling circle amplification were performed, following the manufacturer's protocol (CG000582 Rev E, 10x Genomics). Background fluorescence was chemically quenched. Imaging and signal decoding were done using the Xenium Analyzer instrument (10x Genomics).

### Xenium dataset processing

The 25 datasets included in this study were formatted as anndata using a customized function (https://github.com/Moldia/Xenium_benchmarking) and processed using Scanpy (v1.9.1). To identify main populations in situ, cell-by-gene raw matrices provided by 10x Genomics software were log-transformed and normalized. Neighborhood graphs were then computed considering 40 principal components and 12 neighbors, followed by Leiden clustering. Cell-type annotations were performed using differentially expressed marker genes previously described in Zhang et al.[23]. The same preprocessing steps were performed on cell-by-gene matrices obtained for alternative segmentation methods.

### Annotation of mouse brain architecture

Tissue domains from SRT datasets were manually annotated using the mouse coronal P56 sample from Allen Brain Atlas[49]. Datasets were first processed as described above and then used to create plots displaying cell cluster or spot identity, overlaid on their respective DAPI-, immunofluorescence- or H&E-stained images. Region delimitations were created using enclosed vectorized Bézier curves, which were saved as polygons in scalable vector graphics format (.svg). Annotations were integrated to the cell and/or spot coordinate system from each sample, allowing both classification of cells and/or spots and projection of annotated regions onto the datasets for inspection and downstream analysis (see below). All annotations and detailed instructions on how to use them are freely available at https://github.com/Moldia/Xenium_benchmarking.

### Single-cell RNA-sequencing processing

A subset of the scRNA-seq dataset from Zhang et al.[23] was used in this study. To ensure a fair comparison between the single cell atlas and the various SRT datasets, we included only populations that were spatially mapped in sections of the MERFISH spatially resolved atlas corresponding to the same brain regions represented in the SRT datasets used in this study. Owing to the dimensionality of the dataset, only 20% of the remaining cells were used for comparison, guaranteeing equal representation of all populations. The resulting subset consisted of 193,000 cells used for integration and annotation. Out of those, a subset of 13,800 cells was used as an input for gene imputation. Cell-type signatures for each type were also obtained for the different levels of annotations presented in Zhang et al. and used as an input in cell-typing methods such as SSAM. Region-specific subsets including only populations mapped in the regions presented in Figure 2 were used for comparison with SRT methods.

### Benchmarking Xenium against other SRT methods

To benchmark Xenium against other SRT platforms, DAPI staining and transcript locations were obtained for all datasets. Datasets from open-source technologies were obtained from their original publications (HS-ISS[20], MERFISH[24]), and datasets from commercial platforms were downloaded from the companies web portals (CosMx, Vizgen and Resolved Biosciences). Cells from all datasets were resegmented using common segmentation algorithm Cellpose (cytoplasmic model)[22], followed by the assignment of reads to cells. With the aim of exploring the diffusion of the different technologies, reads identified outside the nuclei were assigned to their closest cell. However, these reads were excluded from the comparison of efficiency and specificity between datasets. To guarantee a fair comparison across platforms, anatomical regions were annotated across datasets, and cells profiled in each region were used for further analyses. For comparison with scRNA-seq, a subset of dataset from Zhang et al.[23] was considered (see previous section). Raw counts were compared between all methods. Only genes present in at least four datasets were used for comparison. Two approaches were used to compare SRT platforms in terms of gene efficiency detection. First, taking scRNA-seq as the reference dataset, the detection efficiency of each gene relative to scRNA-seq was assessed by (1) identifying positive cells (cells with more than one read), (2) computing the median expression of the gene among positive cells and (3) computing the ratio between the two medians. Because we aimed to identify the efficiency of each gene in each SRT method, we divided the SRT median by the scRNA-seq median. Second, to provide a direct comparison between technologies, datasets from all platforms were preprocessed, clustered and annotated. Populations consistently identified across technologies were further used for comparison.

To compare the specificity between technologies, the NCP was computed for each gene. It aims to quantify the presence of coexpressed genes in situ that weren't detected in the single-cell reference dataset. These genes not coexpressed in single-cell but coexpressed in situ can be attributed to non-specific reads, with single-cell datasets serving as the benchmark. In brief, this metric is computed by: (1) identifying pairs of non-coexpressed genes in the reference single cell dataset, (2) quantifying the presence of these pairs across datasets and (3) computing the percentage of these pairs that were not coexpressed in situ. The NCP score ranges from 0 to 1. A score near 0 means most gene pairs not coexpressed in the reference single-cell dataset do coexpress in situ, indicating low specificity. Conversely, a score near 1 indicates that gene pairs not coexpressed in the reference also do not coexpress in situ, reflecting high specificity.

For the side-by-side comparison between Xenium and Visium, manually annotated regions present in both datasets were used. Counts per region were normalized by the total area. The ratio between the number of molecules detected in Xenium and Visium was computed for the common genes.

### Optimal expansion identification

We developed a customized algorithm to identify the optimal expansion distance for each cell type. The rationale behind this algorithm is that the optimal expansion is defined as the point at which the correlation between reads at increasing distances from the nucleus shifts from the nuclear signature to the domain-specific background signature.

To compute this, nuclear expression signatures were defined on the basis of nuclear masks (Fig. 1c), whereas domain-specific signatures were derived from reads not assigned to any cell in Xenium's default expansion. For each cell type–domain pair with at least 5,000 reads, we calculated Pearson's correlation for distance-specific signatures—grouped in 1-μm intervals—with both nuclear and background signatures. The optimal expansion distance was identified as the smallest interval in which correlation with the background exceeded that with the nuclear signature. To determine a cell type's overall mean optimal expansion, we averaged values across all relevant domains. Additionally, we calculated the nuclear edge distance as the mean distance from the cell centroid to the points forming the convex hull of the nuclear region. Similarly, we estimated the cell-edge distance on the basis of all reads assigned to each cell, following Xenium's segmentation and expansion. This algorithm was applied to the mouse brain 1 dataset, as shown in Figure 3a,b.

## Benchmarking segmentation algorithms

We benchmarked the segmentation methods Baysor, MESMER, Watershed, Cellpose and Clustermap against the segmentations provided by 10x Genomics, using uniformly distributed bins across the tissue with a custom pipeline. All segmentation masks were expanded by 1, 2, 5, 10 and 15 μm (0–70 pixels). Watershed segmentation was applied using scikit-image v0.22.0. The DAPI image was first preprocessed as follows: the image was gamma-corrected using default parameters (gamma, 1; gain, 1). Next, the contrast was adjusted using contrast limited adaptive histogram equalization with default values (clip_limit, 0.01; bins, 256), followed by Gaussian blurring with sigma parameter 1 or 3 (referring to 212.5 and 637.5 nm respectively). After retrieving initial cell segments with the watershed algorithm, three postprocessing steps were applied. The first two steps leveraged scikit-image functions to remove small objects with sizes below 64 pixels (13.6 μm), and small holes with a maximum area of 64 pixels (13.6 μm). Owing to local Otsu thresholding, the Watershed segmentation identifies false positive cells in larger regions of low DAPI intensity. Therefore, as a third postprocessing step, a custom background-intensity-based cell filter was applied to remove these cells. The filter removes segmented cells in window sizes of 1,000 × 1,000 pixels that have a mean intensity lower than 0.3 times the mean intensity of the background in the surrounding 2,000 × 2000 pixel window. Baysor (v0.6.2) was tested using scale parameters of 20, 30 and 40 pixels (BA20–BA40); minimum molecules per cell of 30, 50, 70 and 100; and optionally a prior segmentation with prior segmentation confidence of 0.2, 0.4, 0.6, 0.8, 0.9 and 0.99 (for example, CPc BA28). When a prior segmentation was provided, the Baysor algorithm could also automatically adjust the scale parameter. The configurations were run with and without the $z$ coordinate information (that is, 2D versus 3D). The Cellpose (v2.2.3) deep-learning models 'nuclei' (CPn) and 'cyto' (CPc) were applied on the DAPI channels with diameter parameters of none, 20, 30 and 40. Similarly, MESMER was applied with compartment parameters 'nuclear' and 'whole-cell'. Clustermap (github code from 8 November 2022) was applied with Gaussian blurring at sigma values of 1 and 5, and xy_radius 20, 30, 40.

The pipeline takes as input the DAPI image, the gene spots with cell-type annotations and $x$, $y$ and $z$ coordinates, as well as a scRNA-seq reference dataset with cell-type annotations that are matched with the cell-type naming for the spatial cells. The datasets described in 'scRNA-seq processing' were used as references. Cell-type annotations on the nuclear segmentation from 10x Genomics (see 'Xenium dataset processing') was provided. After each segmentation run, a count-per-gene matrix was generated for the cells, and cell types were assigned. With each new segmentation run, new cells were identified, and their cell types were assigned on the basis of the spot's majority vote from the previous nuclear-segmentation-based annotation. For each segmentation output, the pipeline assessed the proportion of assigned reads, number of identified cells, the median and 5th percentile of reads per cell, and the median and 5th percentile of genes per cell. Additionally, we introduced the NMP metric, which is based on the assignment distribution of reads from negative markers: a negative marker for a set of 'negative cell types' is defined, on the basis of the single-cell reference, as a gene that is expressed in less than 0.5% of cells in each of the negative cell types. The NMP measures the percentage of reads of negative markers in the cell types expected to express the gene. To ensure the NMP metric is independent of cell-type proportions, we normalized the mean expressions by the total of the mean expressions of all cell types:

$$\bar{X}_{g,c}^{(m)} = \frac{\bar{x}_{g,c}^{(m)}}{\sum_{c' \in C} \bar{x}_{g,c'}^{(m)}}$$

where $\bar{x}_{g,c}^{(m)}$ is the mean raw expression of a gene ($g$) in a cell type ($c$) and a modality ($m$) (sp, spatial; sc, single-cell reference). $\bar{X}_{g,c}^{(m)}$ can be

understood as the cell-type-balanced reads ratio of a given cell type for a given gene. The normalized mean expression over negative marker–cell type pairs is then:

$$\bar{X}_{neg}^{(m)} = \frac{\sum_{g \in G} \sum_{c \in C_c^{neg}} \bar{X}_{g,c}^{(m)}}{|P^{neg}|}$$

with $G$ representing the set of genes, $C_g^{neg}$ the set of cell types for which gene $g$ is a negative marker and $P^{neg}$ the set of pairs $\{(g, c)\}$, with as $g$ a negative marker for cell type $c$. As we sum the ratios $\bar{X}_{g,c}^{(m)}$, negative markers are equally weighted in principle; however, on the basis of the formulation of $\bar{X}_{neg}^{(m)}$, each negative marker is weighted by its number of negative cell types $|C_g^{neg}|$. Finally, the negative marker purity is calculated as:

$$NMP = \begin{cases} 1 - \left( \bar{X}_{neg}^{(sp)} - \bar{X}_{neg}^{(sc)} \right) & \bar{X}_{neg}^{(sp)} > \bar{X}_{neg}^{(sc)} \\ 1 & otherwise \end{cases}$$

$\bar{X}_{neg}^{(sc)}$ is close to 0. The term is added because the NMP metric aims to measure the extent to which the spatial signal's purity is reduced compared with the scRNA-seq reference. Finally, we scaled the NMP values such that 0 corresponds to the mean NMP of 20 random permutations of cell-type assignments for the binning reference, while the maximum is still 1.

## Simulating Xenium-like datasets and benchmarking preprocessing strategies

To use datasets similar to SRT data while incorporating accurate cell-type annotations, we transformed pre-annotated single-cell datasets in Xenium-like datasets. For this, we first filtered the available datasets provided by CELLxGENE Census[29], keeping only (1) human datasets; (2) chromium v2 datasets; and (3) subsets originated from 1 single donor, to avoid batch effects. Only cell types with more than ten cells were kept. In addition, we identified differentially expressed genes between the remaining cell types using the Scanpy function 'scanpy.tl.rank_genes_groups', keeping the 50 most differentially expressed genes from each cell type to composite the Xenium-like panel later on.

Next, each dataset was transformed to resemble the structure of a Xenium dataset. This transformation includes (1) a reduction in the number of genes detected to match Xenium panels, (2) introduction of non-specific signals, (3) simulation of mis-segmentation and (4) adjustment of the detection efficiency of each gene, mimicking the single cell to Xenium ratios previously detected in mouse brain. To reduce the number of retained genes, marker genes for each cell type were selected from a previously generated list, following the standard approach used in target spatial transcriptomics. The introduction of non-specific signals, simulation of mis-segmented cells and adjustment of the detection efficiency of each gene were implemented to mimic the frequency and characteristics of these effects in Xenium datasets.

Finally, various preprocessing strategies were applied to simulated Xenium-like datasets to identify the preprocessing strategy that best reflected original ground truth clusters. Scanpy and Seurat were used to build the different preprocessing workflows (Extended Data Fig. 6a). In brief, preprocessing workflows consisted of multiple normalization strategies (Pearson's residuals, SCTransform, library-size-based normalization or absence of normalization, defined as 'None' in Extended Data Fig. 6). For library-size-based normalization, different library sizes (10, 100, 1,000 and default) were used as hyperparameters. Next, we optionally performed log-transformation, HVF selection and data scaling. For Pearson's residual normalization, log transformation was always skipped, because it is not required. For SCTransform normalization, both log-transformation and HVF selection steps were skipped, because SCTransform[51] is designed to replace these steps. Subsequently, a graph is constructed, considering the number of principal

components and nearest neighbors as hyperparameters. Finally, clustering was performed using either Leiden, Louvain or Markov cluster algorithms. For each preprocessing workflow tested, the clustering resolution was adjusted to obtain a comparable number of clusters to the reference cell types (±2 clusters). To measure the similarity between both clustering results, different metrics were used, including ARI, VI and FMI. Further details about the simulation are available in the associated Github repository.

### Exploring preprocessing strategies in Xenium datasets

To further assess the effectiveness of the preprocessing strategies on real Xenium datasets, we applied the top-performing preprocessing strategy, identified using simulated data, to the 25 Xenium datasets used in this study (see main text). For the Xenium datasets, default segmentation was used, considering only nuclear reads to avoid the effect of mis-segmentation. To quantify the importance of each of the steps and parameters, we preprocessed each dataset, modifying one step at a time. We quantified the effects of these modifications by measuring the similarity between the new clusters and the ones obtained with the top-performing preprocessing strategy. We used various metrics, including ARI, VI and FMI.

### Benchmarking algorithms to identify spatially variable features

We compared the performance of different algorithms to identify spatially variable features including Sinfonia[34], Squidpy (Moran's I and Geary's C)[30], Giotto (rank and k-means)[36], Seurat (markvariogram and moran's I)[35], hotspot[31], somDE[32], spatialDE[33]. In addition, highly variable genes (HVG) are also identified using Scanpy. Since not all algorithms can be used with full Xenium datasets, we compared the performances of the algorithms in two situations, when possible: usings only 5000 cells and using the entire datasets. All algorithms were applied to a diverse subset of all Xenium datasets available. The algorithm's performance was compared in terms of (1) the ranking of the genes based on how spatially variable they are (2), the number of genes identified as SVF and (3) the similarity between the genes identified as SVF by each algorithm. Finally, since control probes are included in the Xenium experiments which, by definition are not spatially variable, we use this trick to quantify the false positive ratio of each algorithm.

### Benchmarking gene imputation algorithms

We evaluated the performance of seven integration methods (gimVI, SpaGE, Tangram, Liger, Seurat, SpaOTsc and NovoSpaRc) following the published Jupyter notebook by Li et al.[37] (https://github.com/QuKunLab/SpatialBenchmarking/blob/main/BLAST_GenePrediction.ipynb). We used the raw normalized expression matrices for both the scRNA-seq and spatial transcriptomics datasets for input of the integration methods. Because the spatial transcriptomics dataset contained fewer than 300 detected genes, we built the ground truth of our dataset using genes detected in both the spatial transcriptomics and scRNA-seq datasets (total 248 genes). For the evaluation, we used tenfold cross-validation, dividing the genes into ten portions, nine of which were used for training and one for prediction. Because of the large sizes of our datasets (scRNA-seq dataset, ~150,000 cells; spatial transcriptomics dataset, ~80,000 spots), we downsampled the datasets by a factor of ten. This reduction considerably alleviated the computational-resource demands of the integration methods without compromising the results.

We next defined the imputation accuracy of each gene as the PCC between the predicted expression and the real expression. Therefore, values range from 0 to 1, with 1 indicating a perfect prediction. After identifying variability in the imputation accuracy of each gene, we aimed to identify the cause of this variability. For this, we built a linear regression model to predict the imputation accuracy on the basis of the characteristics of each gene, including mean expression, dispersion, subcellular localization and correlation with other genes in the panel.

### Benchmarking domain finder algorithms

Using the adjacent mouse brain slide (slide 1), we evaluated the performance of different domain finder algorithms, including SpaGCN[46], Banksy[44], DeepST[45], STAGATE[48] and SPACEL[47]. The algorithms were used to define tissue domains, adjusting the number of domains identified to match the number of manually annotated tissue domains at different resolutions. In addition, domains were also identified using a primitive neighborhood-based approach, wherein cells were redefined on the basis of the identity of their neighboring cells, forming a cell-by-cell type matrix before clustering. A binning-based approach was also used. These two approaches are simple ways to identify tissue domains and can be used as a baseline to compare the performance of more-sophisticated algorithms. Furthermore, we manually annotated the tissue domains in the slides analyzed using the mouse coronal P56 sample from Allen Brain Atlas[49], which we treated as the ground truth. Several metrics were used to evaluate the performance of each algorithm, including ARI, VI, NMI and FMI.

### Segmentation-free analysis using SSAM

Segmentation-free output was produced using the SSAM package (v1.0.1)[9] with Python v3.6. Two rounds of analysis were performed with different parameterizations.

In the first round, a de novo cell-type-mapping analysis was conducted using the $x$ and $y$ coordinates of Xenium's mouse brain coronal section. SSAM was run at a resolution of 2 pixels $\mu m^{-1}$, with the kernel bandwidth set at 2.5 $\mu m$. Local maximum signal points were sampled from SSAM's vector field at a signal threshold of 0.2, and the signatures of the sampled expression vectors were normalized and clustered by first reducing the data to 40 dimensions using principal component analysis (PCA) and then applying the Leiden algorithm at a resolution of 2.0. Sixty clusters were detected and were assigned a cell type on the basis of a correlation analysis, with the mean gene expression signatures of cells identified in the analysis in Figure 1d. After correlation analysis, a total of 44 cell-type classes were retained. The reduction in class numbers is caused by SSAM's oversegmentation of glial cells and challenges in assigning three gabaergic and three glutamatergic subclusters, which were instead merged into the main GABA/Glut clusters. A SSAM cell-type map was produced and filtered using a median filter to remove potential noise. A scanpy UMAP embedding was generated on SSAM's sampled localmax expression vectors, using an adapted minimum-distance parameter of 0.02.

To assess the $z$-axis coherence of the sample tissue, we devised an algorithm that involved generating a latent space of gene expression signatures through unsupervised analysis. Subsequently, a comparative analysis was conducted on the latent gene expression signatures between the upper and lower halves of the tissue slice.

The construction of the low-dimensional gene expression space employed a segmentation-free de novo approach akin to the SSAM algorithm. A vector field of the gene expression space was established through Gaussian kernel density estimation (KDE) on the mRNA spot signals, using a resolution of 2.5 pixels $\mu m^{-1}$ and a bandwidth of 2.5 $\mu m$. Local maximum locations of the vector field norm were identified and used as sampling locations to construct a gene-expression-by-local-maximum matrix from the vector field. This matrix was subjected to PCA, retaining the first $n$ principal components explaining over 80% of the total variance. The resulting low-dimensional gene expression space was utilized to create two vector fields for the gene expression space, maintaining the previously mentioned parameters. The division into top and bottom halves of the tissue slice was achieved by binning molecules into a 2-$\mu m$ grid in the $x$ and $y$ directions. The center of gravity of molecules in the $z$-direction (mean of $z$-coordinate values) determines the assignment of molecules to the 'top' or 'bottom' halves. A KDE was then applied to both halves, with parameters consistent with those previously mentioned. Subsequently, the resulting vector fields underwent transformation into latent space using the PCA model. The cosine similarity between the

two latent space vector fields served as a metric for assessing *z*-axis incoherence. Analogous to the SSAM algorithm, vector field regions with a total expression norm below 5 were considered low-confidence and were excluded from subsequent analyses.

Visualization of the incoherence maps employed the heatmap visualization function from Matplotlib, wherein color values represented signal coherence, and alpha values denoted local molecular density (measured by vector field norm). Regions classified as low-confidence were excluded from the visual representation.

Moreover, areas exhibiting significant coherence were selectively chosen for display, using a SSAM-inspired pipeline. A supervised, 3D execution of the SSAM algorithm assigned cell types to all molecules within the visualization area. Molecules were then depicted as particle clouds in 3D space, colored according to their assigned cell types, following the project's overarching cell-type coloring scheme. For individual samples, a coherence score of <0.2 proved to be strongly indicative of overlapping cell-type structures. The mean incoherence score inside the area of the DAPI-based nucleus segmentation served as a measure of local signal coherence, and the ratio of nuclei with a mean signal coherence of <0.2 was indicative of strong overlap.

### Points2Regions

Points2Regions[10] was used as one of the segmentation-free approaches. In essence, Points2Regions is a plugin for TissUUmaps 3[52], intended for quick exploratory and interactive dissection of molecular patterns in in situ transcriptomics data. At its core, the plugin works by collecting markers in spatial bins of width $w$. Each bin thus comprises a composition of molecular markers. Adjacent bins are blurred together using a Gaussian filter parameterized by the standard deviation $\sigma$. Bins containing few markers are excluded based on a user-defined threshold $\tau$. Bins passing the threshold are finally normalized by total count and clustered using mini-batch KMeans clustering with $k$ clusters. The plugin thus comes with four tunable parameters: $w$, $\sigma$, $\tau$ and $k$, each respectively set here to 1 μm, 3 μm, 0 and 100 for all experiments.

### Reporting summary

Further information on research design is available in the Nature Portfolio Reporting Summary linked to this article.

### Data availability

Three types of Xenium datasets were used through the manuscript, including (1) datasets provided by 10x Genomics, (2) datasets published elsewhere and (3) datasets generated specifically for this project. First, for the 10x Genomics datasets, the original datasets used in this study can be obtained from in https://www.10xgenomics.com/datasets (accessed 5 March 2024). In addition, the spinal-cord datasets used were originally published by Kukanja & Mattsson-Langseth et al.[53]. Freshly generated datasets include four mouse-brain sections, labeled as 'hm' through the study. Their original data can be downloaded from: https://doi.org/10.5281/zenodo.10566172 (ref. 54).
In addition, we have also made Xenium datasets available as AnnData objects. These files can be downloaded from various Zenodo repositories: https://doi.org/10.5281/zenodo.11124988 (ref. 55), https://doi.org/10.5281/zenodo.11121221 (ref. 56) and https://doi.org/10.5281/zenodo.11120307 (ref. 57).
All datasets used in the comparison between SRT platforms (Fig. 2) are publicly available. Commercial platforms provide datasets through their respective data portals: MERSCOPE, https://vizgen.com/data-release-program/; CosMx, https://nanostring.com/products/cosmx-spatial-molecular-imager-ffpe-dataset/; and Molecular Cartography, https://resolvebiosciences.com/datasets/. For both MERFISH and HS-ISS, datasets are available in their original publications[20,24]. In addition, resegmented and regionally annotated datasets that can be used to reproduce the comparison can be found at https://doi.org/10.5281/zenodo.11619309 (ref. 58).

### Code availability

All the code used in this analysis can be found at https://github.com/Moldia/Xenium_benchmarking. Since different tools require their own environment, analysis is subdivided in folders, providing different conda recipes files to recreate the environments needed to reproduce the analysis.

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

### Acknowledgements

The datasets generated using Xenium and described in this manuscript provided by 10x Genomics correspond to data generated using an in-development gene panel, chemistry and instrument. Thus, they might not be indicative of final application performance. We thank the ISS facility in Scilifelab, Stockholm, for generating the mouse brain datasets used throughout the manuscript. We would like to thank the organizers and participants of the de.NBI BioHackathon SpaceHack project for their contributions to the development of the algorithms and benchmarking pipelines used throughout the study.

This work was supported by grants from the Knut and Alice Wallenberg Foundation (KAW 2018.0172), the Erling Persson Foundation, the Chan Zuckerberg Initiative (SVCF 2017-173964), Cancerfonden (MN: CAN 2018/604) and the Swedish Research Council (MN: 2019-01238). N.I. is financially supported by the European Commission EU Horizon 2020 research and innovation program. P.C. is financially supported by the Knut and Alice Wallenberg Foundation as part of the National Bioinformatics Infrastructure Sweden at SciLifeLab. Part of the computations and data storage was enabled by resources provided by the Swedish National Infrastructure for Computing (SNIC) at UPPMAX, partially funded by the Swedish Research Council through grant agreement no. 2018-05973. S.T. is financially supported by the Federal Ministry of Education and Research of Germany in the framework of SAGE (FKZ: 031L0265).

### Author contributions

S.M.S. designed the study. S.M.S. obtained, formatted and described the Xenium datasets used in this study. C.M.L. and S.M.S. compared the Xenium datasets against other spatial technologies. S.M.S., P.C., M.C. and K.T. annotated the cell populations. P.C. and S.M.S. annotated the tissue domains in all datasets and S.M.S. benchmarked tissue

domain identification algorithms. M.G. and S.M.S. conceptualized and implemented the workflow used to compare preprocessing strategies, both on simulated and on real Xenium data. M.D.L., L.B.K. and H.R. contributed to benchmarking segmentation strategies. C.W., A.A., N.I., S.T. and C.A. applied segmentation-free algorithms to Xenium datasets, and N.I., S.T. and S.M.S. evaluated the patterns identified. S.H. and T.H. adapted and implemented the benchmarking pipeline for gene imputation. N.R. assembled the code repository and documentation. S.M.S., C.A. and A.A. contributed to data analysis while S.M.S., M.N., N.I., P.C., F.J.T. and M.D.L. contributed to data interpretation. S.M.S., N.I., M.N., P.C., M.D.L., C.A., C.W. and A.A. contributed to writing the manuscript. All authors read and approved the content of this manuscript.

## Funding

## Competing interests

M.N. was an advisor for 10x Genomics when this manuscript was initially submitted, but he is no longer involved in any advisory role for the company. F.J.T. consults for Immunai, Singularity Bio, CytoReason and Omniscope, and has ownership interest in Dermagnostix and Cellarity. M.D.L. contracted for the Chan Zuckerberg Initiative and received speaker fees from Pfizer and Janssen Pharmaceuticals. S.M.S., C.M.L. and M.G. are co-founders of Spatialist, a data-analysis company focused on spatial omics. The remaining authors declare no competing interests.

## Additional information

**Extended data** is available for this paper at https://doi.org/10.1038/s41592-025-02617-2.

**Correspondence and requests for materials** should be addressed to Sergio Marco Salas or Mats Nilsson.

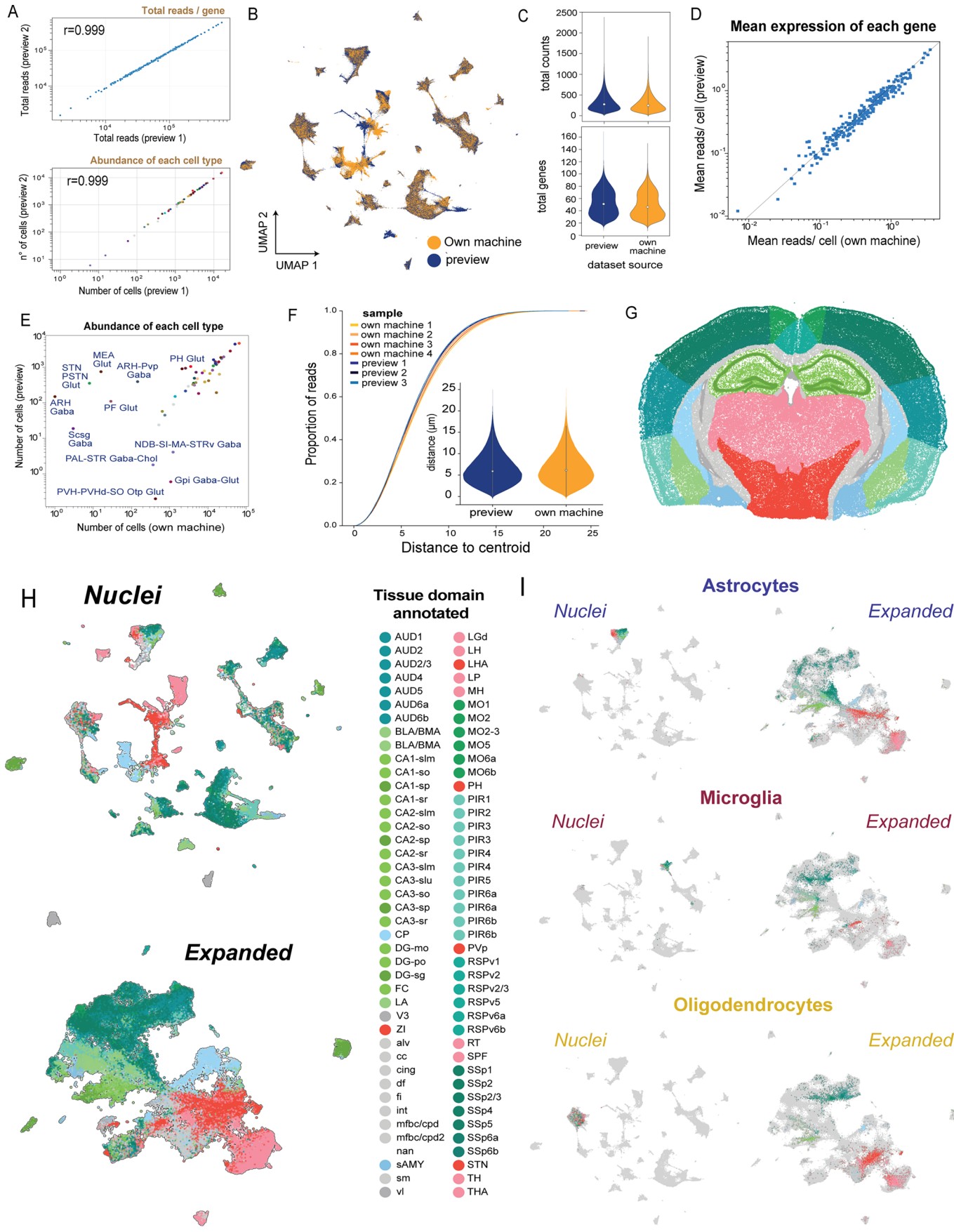

**Extended Data Fig. 1 | See next page for caption.**

**Extended Data Fig. 1 | Extended analysis of the mouse brain datasets.**
**a**. Comparison between two Xenium biological replicates, including a scatter plot representing the total transcripts of each gene in the preview 1 dataset compared to the total transcripts of each gene in preview dataset 2 (up), with axes represented in log10 scale. In addition, a scatter plot representing the abundance of each cell type identified in the preview dataset 1 compared to the preview dataset 2 is included (down). Axes are represented in log10 scale **b**. UMAP representation of cells from the 7 mouse brain datasets, colored by the experiment of origin. **c**. Violin plot representing the transcripts/cell (up) and genes/cell (bottom) identified on each of the mouse brain datasets. Violins are colored by the experiment of origin. **d**. Scatter plot representing the mean transcripts/cell of each gene in the home made datasets compared to their mean transcripts/cell on the preview datasets. Axes are represented in log10 scale. **e**. Scatter plot representing the abundance of each cell type identified in the home made datasets compared to the preview datasets. Axes are represented in log10 scale. **f**. Density plot illustrating the density of reads depending on their distance to their assigned cell centroid. Individual lines represent different samples. A violin plot quantifying the subcellular distribution of reads experiment is included (bottom, right). **g**. Spatial map of the mouse brain section 1, colored by annotated tissue domains **h**. UMAP representations of the cells analyzed from the 7 mouse brain datasets obtained from using only nuclear information (up) or expanded segmentation masks (bottom) to assign reads to individual cells. Cells are colored by the annotated tissue domain where they are found. **i**. UMAP representations of specific populations including astrocytes (up), microglia (middle) and oligodendrocytes (bottom) defined using nuclei-based segmentation masks (left) and projected onto the cells' expanded segmentation masks (right). Cells are colored by the tissue domain where they are found, according to panel G.

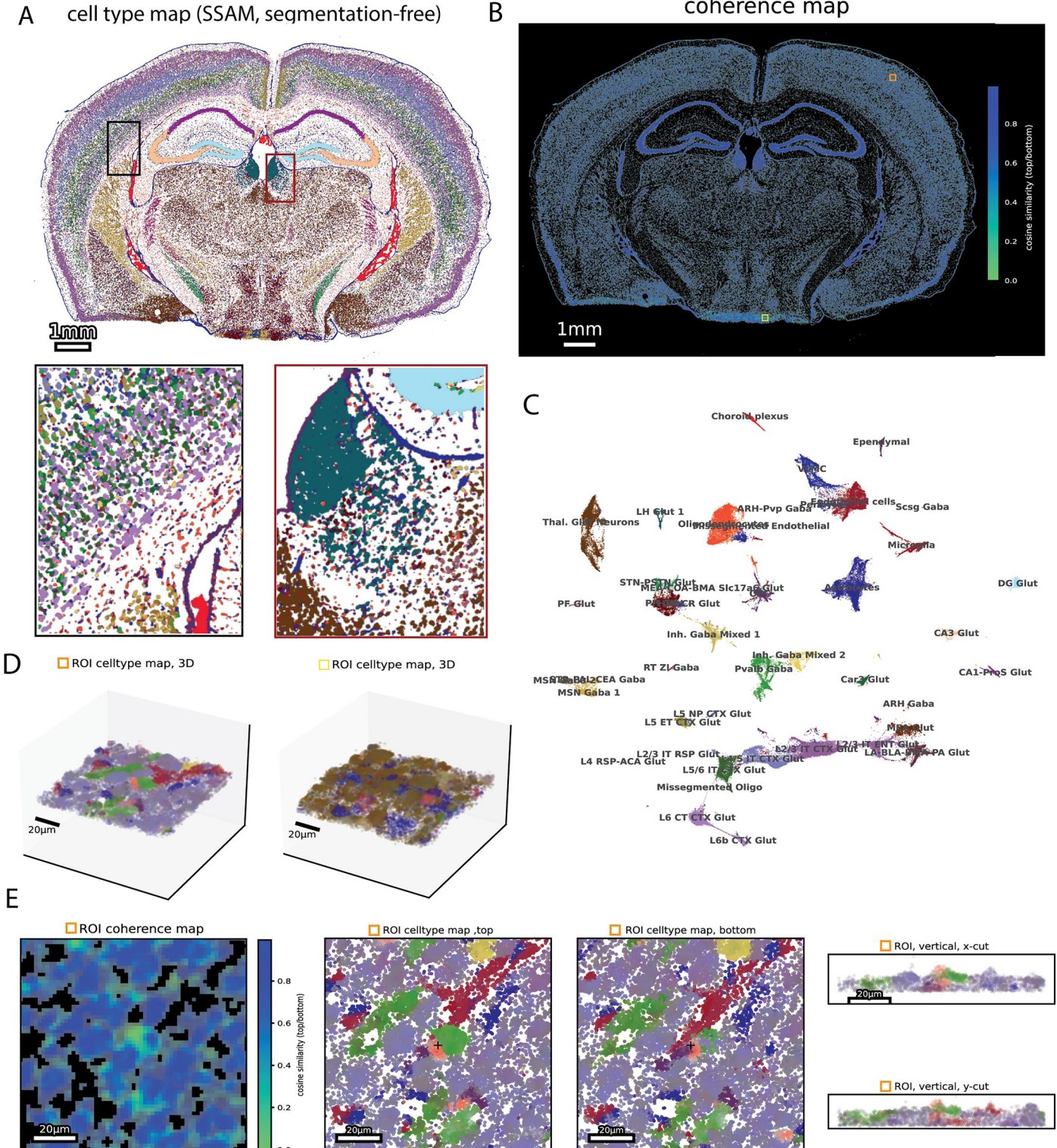

**Extended Data Fig. 2 | SSAM segmentation-free analysis of Xenium datasets.**
**a**. Spatial map of the clusters obtained from the SSAM segmentation-free analysis applied on the mouse brain section 2. Reads are colored following the colormap used on Fig. 1c to represent cell types. Regions of interest are highlighted in the whole map (up) and visualized in the bottom part of the panel. **b**. 3D-coherence map of the dataset analyzed in panel A. Colors represent the cosine similarity, with low values representing regions with a low top-bottom signal coherence, indicating potentially overlapping cells. ROIs selected as examples of regions of low 3D-coherence are indicated using colored squares **c**. Umap of the signatures

identified by SSAM analysis included in mouse brain section 2, colored by the cell types represented in Extended Data Fig. 2a **d**. 3-D visualization of the ROIs with low 3D-coherence regions, as indicated in panel B. **e**. Spatial maps illustrating the 3D nature of Xenium datasets. A second region of interest with a low 3D-coherence is presented, indicating the presence of overlapping cell types. The coherence map (left) is complemented by X- and Y-axis (XY maps) of the ROI, viewed from the bottom and top (mid) and XZ and YZ maps (right). Each spot represents an individual read, colored based on the cell type assignment done using SSAM and following the color code presented in Fig. 1c.

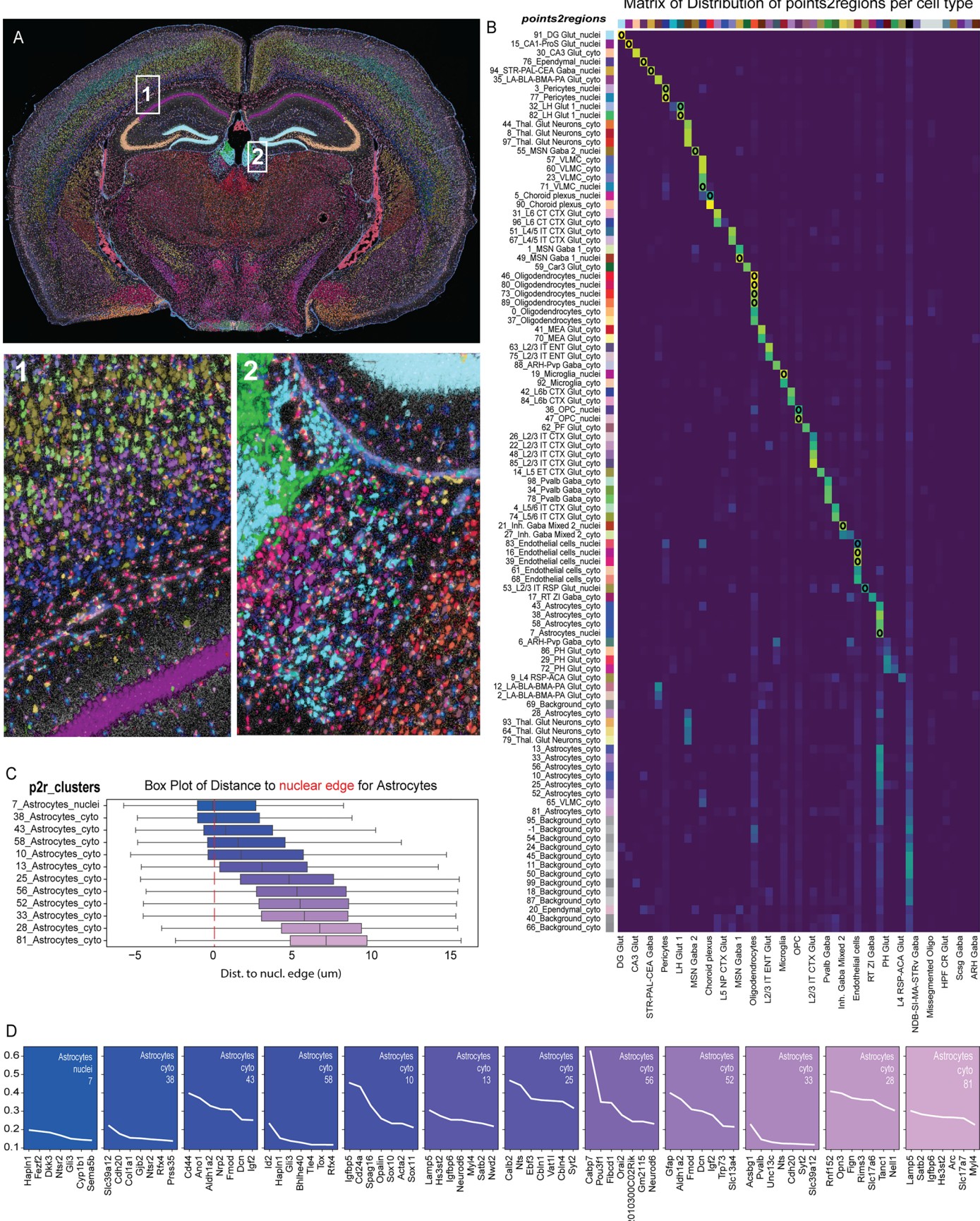

**Extended Data Fig. 3 | See next page for caption.**

**Extended Data Fig. 3 | Subcellular analysis of Xenium datasets with Points2Regions. a**. Spatial map of the entire mouse brain 2 dataset (up), with reads colored by their Points2Regions cluster. The color map used is shown in panel B. Two regions of interest (1,2) are highlighted in the entire map and visualized (bottom). **b**. Confusion matrix between the Points2Regions clusters and the segmentation-based clusters represented in Fig. 1c. Points2Regions clusters are annotated based on (1) a number (2), the cell type its reads had been mostly assigned to in the segmentation-based analysis and (3) the main subcellular localization of the reads assigned to each cluster (cyto or nuclei). Points2Regions clusters that have a majority of their reads within the nuclei boundaries, defined using DAPI staining, are annotated as nuclei clusters. Oppositely, clusters with most of their reads outside the nuclei boundaries are annotated as cytoplasmic clusters (cyto). On the other hand, cytoplasmic clusters (cyto) present most of their reads outside the segmented nuclei.

Nuclear clusters are represented by the presence of a circle in the confusion matrix. This circle is placed, for the rows where it's needed, in the cell with the highest value of the row, with the highest similarity between a segmentation-based cluster and the Points2Regions' cluster represented on the row. **c**. Box plot representing the distribution of reads assigned to the Points2Regions clusters in astrocytes in relation to their distance to the nuclei edge. Red horizontal dashed line at y=0 represents the nuclear edge. Box plot represents percentiles excluding outliers (0,25,50,75,100), with center representing median distance. **d**. Differentially expressed genes for each subcellular cluster found in astrocytes. Y-axis represents the relative percentage of reads of a certain gene assigned to the interrogated cluster. Note that this relative percentage is computed considering only the astrocytic clusters, meaning that, overall, the sum of all percentages in all astrocytic clusters should sum 1.

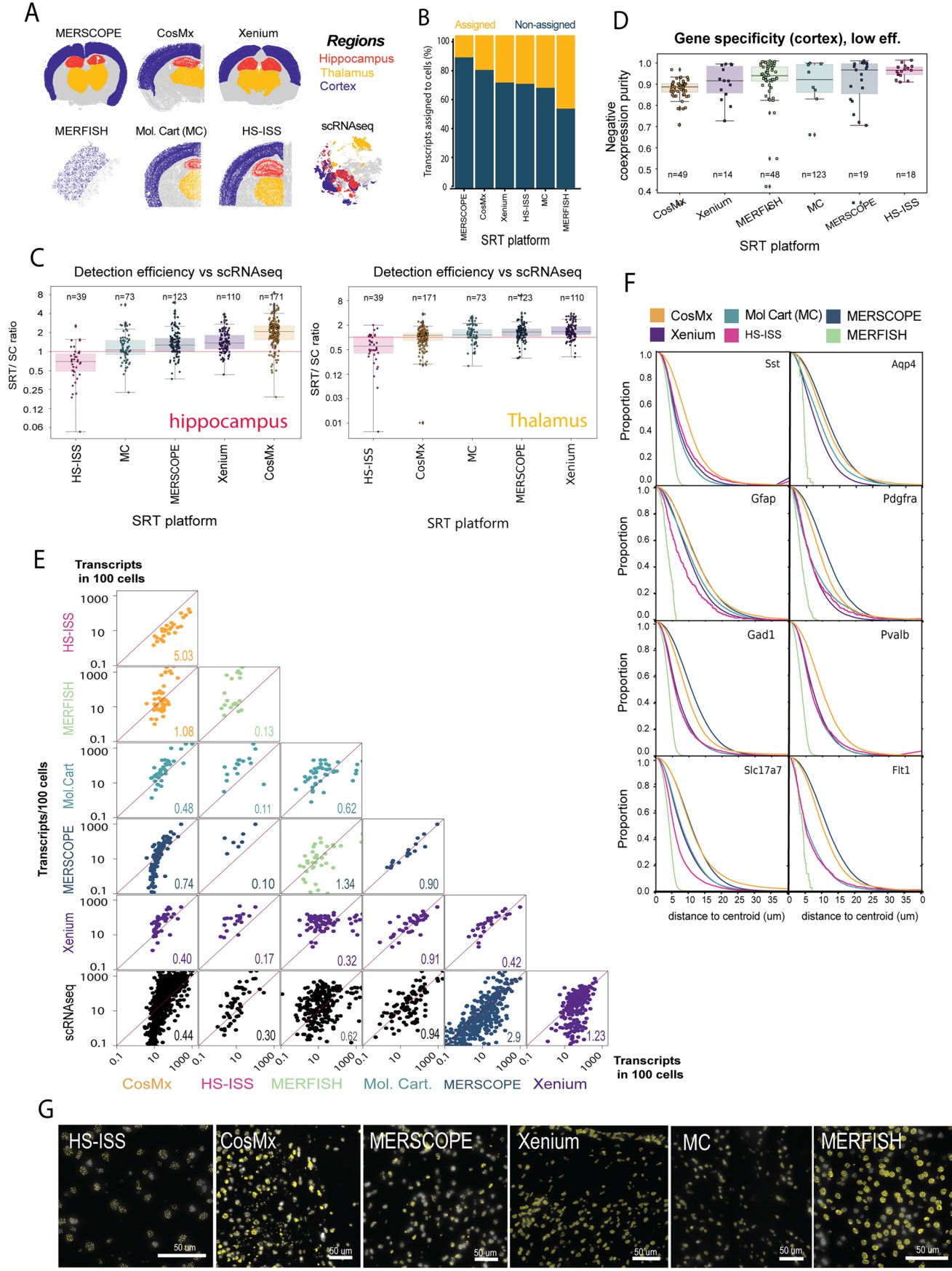

**Extended Data Fig. 4 | See next page for caption.**

**Extended Data Fig. 4 | Comparison of Xenium with SRT platforms. a.** Spatial map of SRT datasets, colored by region. The scRNA-seq dataset is represented as a UMAP colored by region of origin. **b.** Stacked bar plot representing the percentage of transcripts assigned to cells in each datasets after Cellpose. **c.** SRT/scRNA-seq gene efficiency ratios of different SRT methods in the hippocampal (left) and thalamic (right) regions. Boxplots represent the distribution of the efficiencies, divided in quartiles, where the central line represents the median efficiency. Gene ratios are represented as individual dots. **d.** Box plot representing the negative coexpression purity (NCP) of each SRT method, including only genes with an efficiency ratio below 1. Boxplots represent the distribution of the NCP scores, divided in quartiles, where the central line represents the median NCP. NCP scores are represented as individual dots for each method. **e.** Pairwise comparison of the detection efficiency between each SRT and scRNA-seq dataset in the cortical region. For each pair, a scatter plot of the number of transcripts detected per gene in SRT method 1 (y-axis) and SRT method 2 (x-axis) is included. Only common genes are included in the comparison. Red line represents x=y. The median of ratios for each pair of methods is included (bottom,right). Spots of each subplot are colored by the method that presented a higher median in each comparison. **f.** Density plots illustrate the cumulative proportion of reads depending on their distance from the centroid for individual genes across technologies. Values indicate the proportions of reads that are found at distances greater than or equal to the specified distance. **g.** Regions of interest corresponding to resegmented datasets across platforms. DAPI staining is shown as a background and reads assigned to resegmented cells are overlaid as yellow dots.

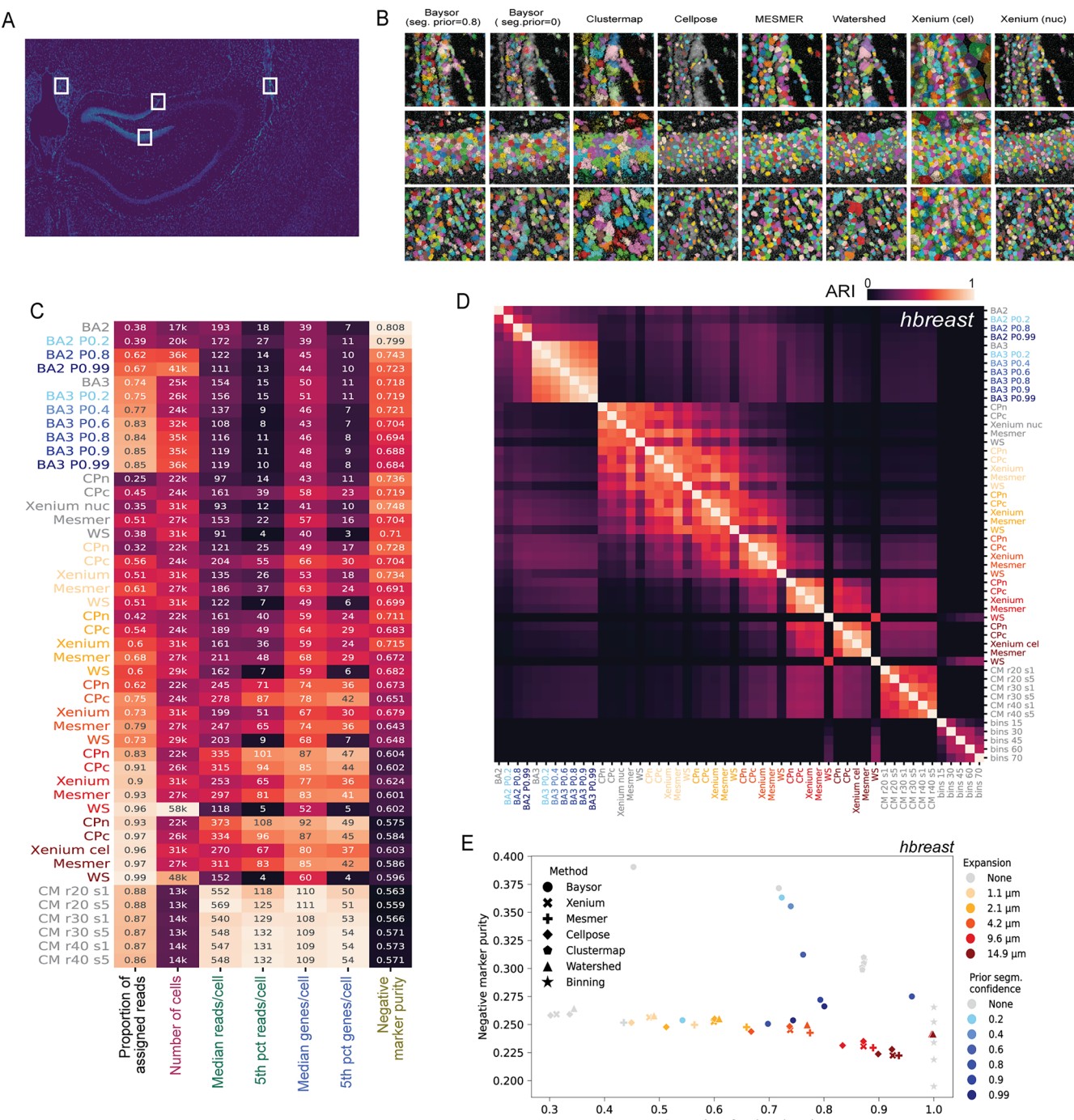

**Extended Data Fig. 5 | Extended benchmarking of segmentation strategies.**
**a** Localization of regions of interest represented in Extended Data Fig. 5b and
Fig. 3c. **b**. Regions of interest representing the cells identified using different
segmentation algorithms in a region of interest outlined in Extended Data 5B.
DAPI background is represented as a background and individual isolated color-
specific masks represent individual cells. Segmentation strategies were selected
to represent different segmentation outputs, as described in Fig. 3c. Each ROI
represents an area of 160 × 160 μm. **c**. Heat map representing the segmentation
metrics of all segmentation strategies described in Fig. 3d. **d**. Adjusted rand
index (ARI) between the different outputs produced by combinations of
segmentation algorithms, hyperparameters and expansions when applied to
human breast sections. Segmentation methods included Cellpose

(CPn: nuclei, CPc: cyto models), binning (bins), clustermap (CM), watershed
(WA), Mesmer, Baysor (BA) and Baysor with prior segmentation (Baysor Px.x).
Xenium segmentation were also included in the comparison (XENIUM cel,
XENIUM nuc). Hyperparameters for each method are described in methods.
Methods on the y-axis were colored depending on the expansion performed after
segmentation. 315 evaluated configurations of the grid search were reduced
to the shown 52 top performers per hyperparameter group (highest negative
marker purity **e**. Scatter plot representing the number of reads assigned (x-axis)
and the negative marker purity (y-axis) of different assessed segmentation
strategies in human breast tumor samples. The name and color of each
segmentation strategy are represented as in Fig. 3d.

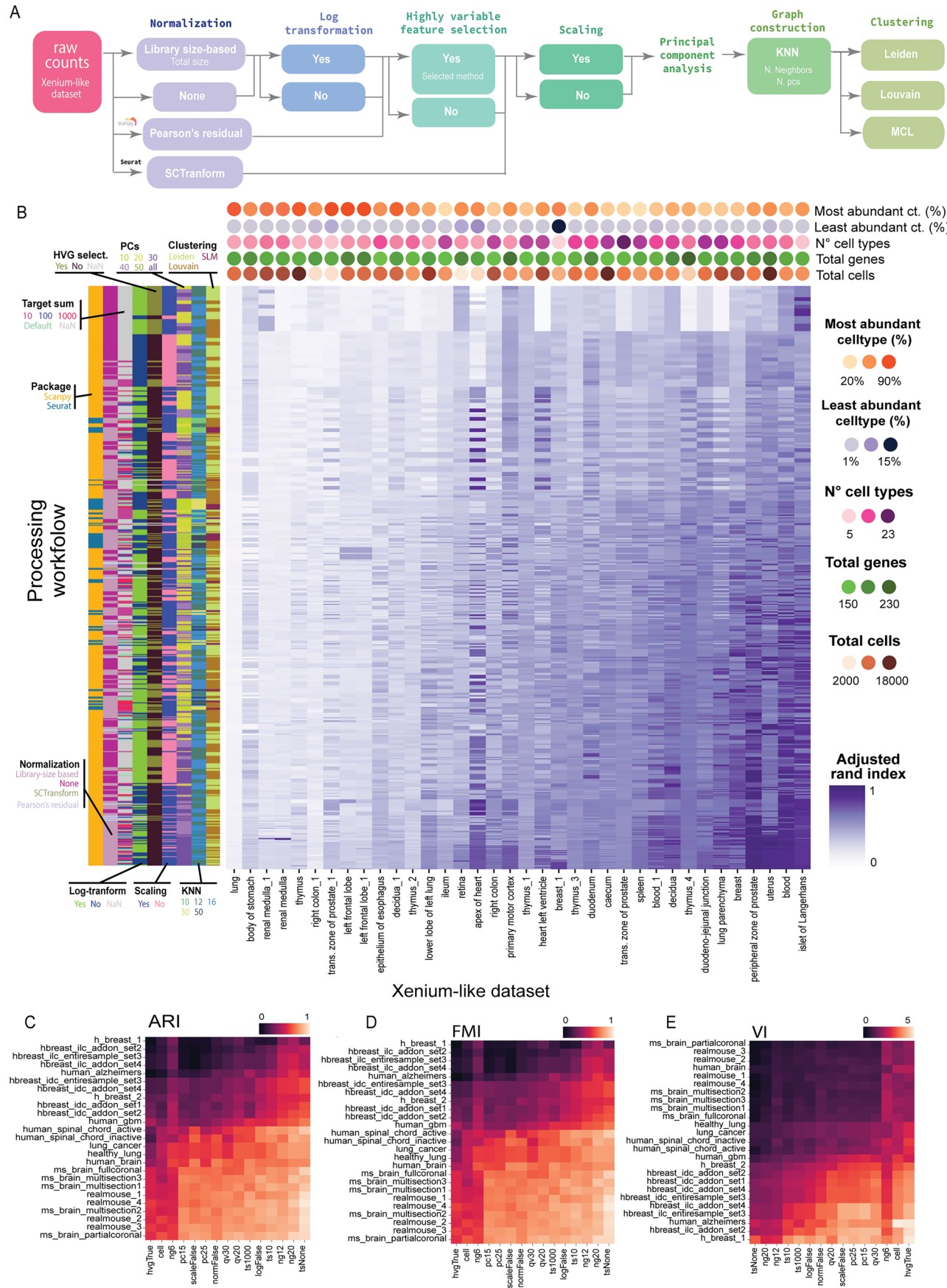

**Extended Data Fig. 6 | See next page for caption.**

**Extended Data Fig. 6 | Extended analysis on preprocessing. a**. Workflow of the different preprocessing steps and parameters considered in the assessment of the best preprocessing workflow. **b**. Heat map representing the Adjusted Rand Index (ARI) between the clusters derived from the different preprocessing workflows and the ground truth cell type labels. Preprocessing workflows are sorted from best (bottom) to worst (top) based on their median ARI. Datasets are also sorted based on their median ARI, indicating in which datasets it was possible to recover the original cell type labels better (right) and in which ones it was more difficult to achieve (left). A summary of the processing setups is summarized on the left part of the panel, with every row representing a specific step in the preprocessing workflow and every color representing the specific hyperparameter/algorithm chosen. In addition, specific characteristics of the simulated datasets are included on the top part of the panel in the form of dot plot **c**. Heat map representing the mean Adjusted Rand Index (ARI)

between the clusters obtained when applying the most optimal preprocessing workflow (identified in Fig. 4c) to different Xenium datasets and the clusters obtained with the same workflow, but modifying different preprocessing steps, specified in the x-axis, in the different Xenium datasets (y-axis). A reduced ARI signifies decreased similarity between clustering outputs, highlighting a more pronounced impact on the workflow when altering a specific parameter. **d**. Same as C, but using Fowlkes-Mallows Index (FMI) to measure the similarity between the clustering outputs. A low FMI indicates differences between the clustering outputs, suggesting a more pronounced impact on the workflow when altering a specific parameter. **e**. Same as C, but using the variability index (VI) to measure the similarity between the clustering outputs. A high VI indicates important differences between the clustering outputs, suggesting a more pronounced impact on the workflow when altering a specific parameter.

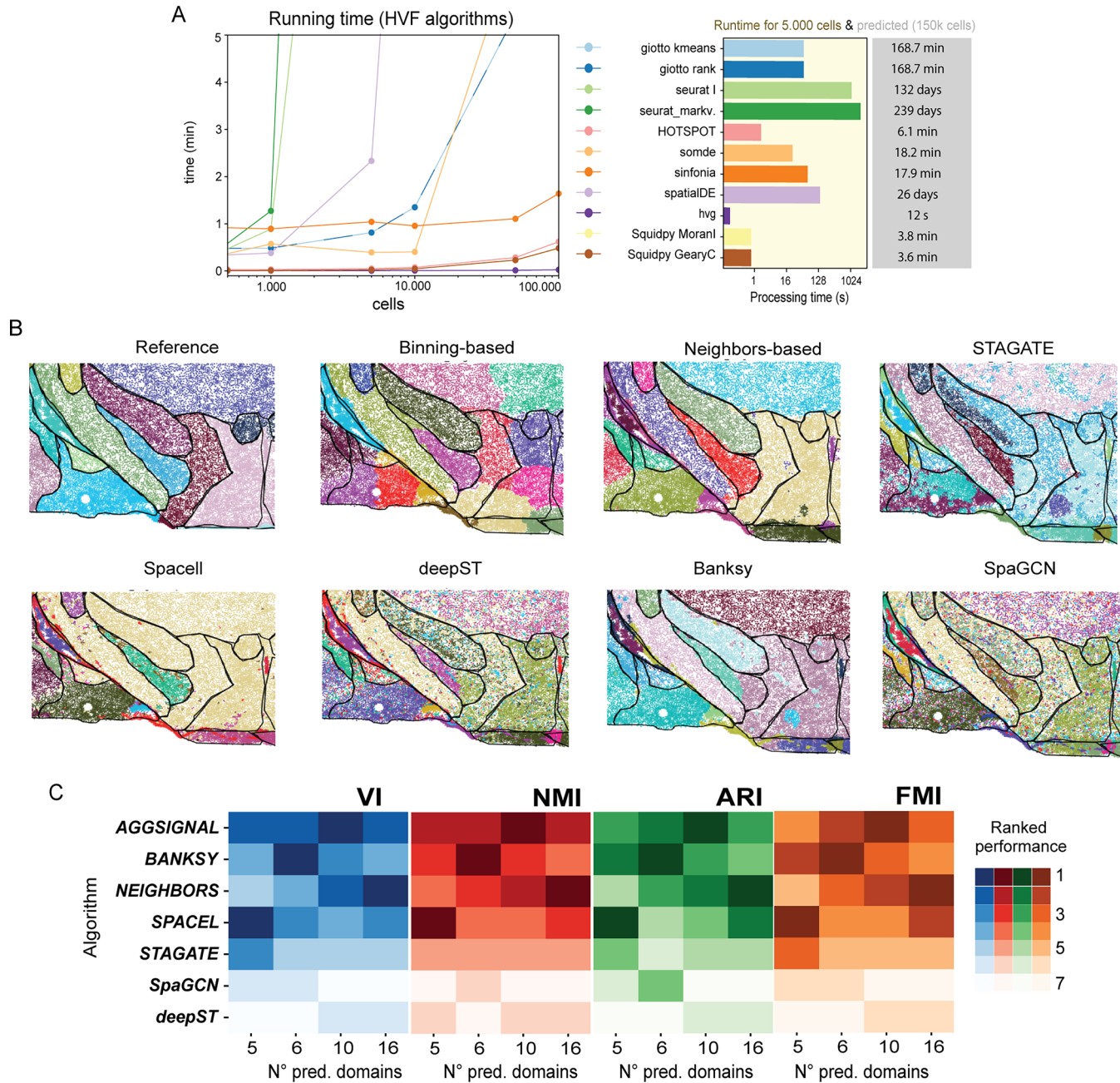

**Extended Data Fig. 7 | Extended exploration of the SVF identification algorithms. a.** Running time of the different algorithms used to identify SVFs. The running times depending on the number of cells used as an input are shown as a line plot (left), together with a bar plot representing the processing times of different algorithms when using 5.000 cells (middle) and the predicted running times for each algorithm when using a full dataset (~150.000 cells) (right). **b.** Spatial map of the manually annotated domains identified in the mouse brain section (ROI2) (replicate 1, left) and the domains identified by different algorithms. **c.** Ranked performance of different algorithms in identifying tissue domains in mouse brain sections (ROI2), using the manually segmented domains as a reference. Four metrics are used: Adjusted Rand index (ARI), variability index (VI), NMI and Fowlkes-Mallows Index (FMI). Different numbers of domains are predicted, based on the number of domains included in the hierarchical annotation of the tissue done manually[6,9,17,49].

# Reporting Summary

## Statistics

For all statistical analyses, confirm that the following items are present in the figure legend, table legend, main text, or Methods section.

| n/a | Confirmed | |
|---|---|---|
| ☐ | ☒ | The exact sample size (*n*) for each experimental group/condition, given as a discrete number and unit of measurement |
| ☐ | ☒ | A statement on whether measurements were taken from distinct samples or whether the same sample was measured repeatedly |
| ☒ | ☐ | The statistical test(s) used AND whether they are one- or two-sided<br>*Only common tests should be described solely by name; describe more complex techniques in the Methods section.* |
| ☐ | ☒ | A description of all covariates tested |
| ☐ | ☒ | A description of any assumptions or corrections, such as tests of normality and adjustment for multiple comparisons |
| ☐ | ☒ | A full description of the statistical parameters including central tendency (e.g. means) or other basic estimates (e.g. regression coefficient) AND variation (e.g. standard deviation) or associated estimates of uncertainty (e.g. confidence intervals) |
| ☒ | ☐ | For null hypothesis testing, the test statistic (e.g. *F*, *t*, *r*) with confidence intervals, effect sizes, degrees of freedom and *P* value noted<br>*Give P values as exact values whenever suitable.* |
| ☒ | ☐ | For Bayesian analysis, information on the choice of priors and Markov chain Monte Carlo settings |
| ☐ | ☒ | For hierarchical and complex designs, identification of the appropriate level for tests and full reporting of outcomes |
| ☐ | ☒ | Estimates of effect sizes (e.g. Cohen's *d*, Pearson's *r*), indicating how they were calculated |

*Our web collection on statistics for biologists contains articles on many of the points above.*

## Software and code

Policy information about availability of computer code

| Data collection | Xenium datasets were either made available by 10X Genomics or collected using 10X Xenium instruments. Datasets were preprocessed using the code included in https://github.com/Moldia/Xenium_benchmarking v1.2.0 |
|---|---|
| Data analysis | All the code used in this study can be found at https://github.com/Moldia/Xenium_benchmarking v1.2.0<br><br>Python packages used through the study include:<br>affine==2.4.0, anndata==0.8.0, alphashape==1.3.1, biopython==1.81, click==8.1.5, click-log==0.4.0, click-plugins==1.1.1, cloudpickle==2.1.0, contextily==1.3.0, cython==3.0.2, dask==2022.2.0, dask-image==2021.12.0, descartes==1.1.0, doubletdetection==4.2, fiona==1.9.5, geographiclib==2.0, geopandas==0.10.2, geopy==2.4.0, gprofiler-official==1.0.0, h5py==3.7.0, holoviews==1.16.2, igraph==0.9.11, imagecodecs==2021.11.20, imageio==2.21.0, leidenalg==0.8.10, libpysal==4.7.0, louvain==0.7.2, matplotlib==3.5.2, matplotlib-scalebar==0.8.1, matplotlib-venn==0.11.9, mygene==3.2.2, naivede==1.2.0, networkx==2.6.3, numba==0.56.0, numpy==1.21.6, omnipath==1.0.5, pandas==1.3.5, phenograph==1.5.7, pooch==1.7.0, pydantic==1.9.1, pynndescent==0.5.7, pyparsing==3.0.9, pyproj==3.2.1, rasterio==1.2.10, rtree==1.0.1, scanpy==1.9.1, scikit-image==0.19.3, scikit-learn==1.0.2, scipy==1.7.3, seaborn==0.11.2, shapely==2.0.1, spatialde==1.1.3, spatialdm==0.1.0, squidpy==1.2.2, statsmodels==0.13.2, tifffile==2021.11.2, toolz==0.12.0, tqdm==4.64.0, umap-learn==0.5.3, xarray==0.20.2, zarr==2.12.0<br><br>Furthermore, R packages include:<br>Seurat(4.3.0), SeuratObject( 4.1.3), Giotto (1.1.2) |

For manuscripts utilizing custom algorithms or software that are central to the research but not yet described in published literature, software must be made available to editors and reviewers. We strongly encourage code deposition in a community repository (e.g. GitHub). See the Nature Portfolio guidelines for submitting code & software for further information.

## Data

Policy information about availability of data

All manuscripts must include a data availability statement. This statement should provide the following information, where applicable:
- Accession codes, unique identifiers, or web links for publicly available datasets
- A description of any restrictions on data availability
- For clinical datasets or third party data, please ensure that the statement adheres to our policy

Three types of Xenium datasets were used through the manuscript, including (1) datasets provided by 10X Genomics, (2) datasets published elsewhere and (3) datasets generated specifically for this project. First, for the 10X Genomics datasets, the original datasets used in this study can be obtained from in https:// www.10xgenomics.com/datasets [03.05.2024]. In addition, the spinal cord datasets used were originally published by Kukanja & Mattsson-Langseth et al. (24). Lastly, freshly generated datasets include four mouse brain sections, labelled as "hm" through the study. Their original data can be downloaded from: https:// doi.org/10.5281/zenodo.10566172.
 In addition, we have also made Xenium datasets available as AnnData objects. These files can be downloaded from various¬¬ Zenodo repositories (https:// doi.org/10.5281/zenodo.11124988 , https://doi.org/10.5281/zenodo.11121221 , https://doi.org/10.5281/zenodo.11120307 )
All datasets used in the comparison between SRT platforms (Figure 2) are publicly available datasets. In the case of the commercial platforms , the datasets are available in the company's data portals (MERSCOPE: https://vizgen.com/data-release-program/, CosMx: https://nanostring.com/products/cosmx-spatial-molecular-imager/ffpe-dataset/, Molecular Cartography: https://resolvebiosciences.com/datasets/). For both MERFISH and HS-ISS, datasets were made available in their original publications (3,4). In addition, resegmented and regionally annotated datasets, ready to reproduce the comparison, can be found at https:// doi.org/10.5281/zenodo.11619309.
Allen brain atlas single cell dataset ABC atlas) used through the study is available at https://portal.brain-map.org/atlases-and-data/bkp/abc-atlas.

## Human research participants

Policy information about studies involving human research participants and Sex and Gender in Research.

| Reporting on sex and gender | N/A |
| --- | --- |
| Population characteristics | N/A |
| Recruitment | N/A |
| Ethics oversight | N/A |

Note that full information on the approval of the study protocol must also be provided in the manuscript.

# Field-specific reporting

Please select the one below that is the best fit for your research. If you are not sure, read the appropriate sections before making your selection.

☒ Life sciences        ☐ Behavioural & social sciences        ☐ Ecological, evolutionary & environmental sciences

For a reference copy of the document with all sections, see nature.com/documents/nr-reporting-summary-flat.pdf

# Life sciences study design

All studies must disclose on these points even when the disclosure is negative.

| Sample size | All publicly and internally available Xenium datasets at the time of the study (25 different Xenium datasets) were used with the aim of generalize the conclusions of the study. Due to the recent commercialization of the Xenium product, limited datasets are available |
| --- | --- |
| Data exclusions | Cells that did not pass several quality filters based on the number of transcripts and genes detected were excluded from analysis |
| Replication | Due to the recent commercialization of the Xenium product, limited datasets were available through the study. As a consequence,  no biological replicates could be used. Mouse brain datasets, however, presented technical replicates,which were used through the study to assess the robustness of the method. |
| Randomization | This study presents reduce sample size with n=1 in many cases. Thus, randomization is not relevant to this study |
| Blinding | Due to the limited sample size, with n=1 in most cases, blinding does not apply to this study. |

# Reporting for specific materials, systems and methods

We require information from authors about some types of materials, experimental systems and methods used in many studies. Here, indicate whether each material, system or method listed is relevant to your study. If you are not sure if a list item applies to your research, read the appropriate section before selecting a response.

## Materials & experimental systems

| n/a | Involved in the study |
|-----|-----------------------|
| ☒ | ☐ Antibodies |
| ☒ | ☐ Eukaryotic cell lines |
| ☒ | ☐ Palaeontology and archaeology |
| ☒ | ☐ Animals and other organisms |
| ☒ | ☐ Clinical data |
| ☒ | ☐ Dual use research of concern |

## Methods

| n/a | Involved in the study |
|-----|-----------------------|
| ☒ | ☐ ChIP-seq |
| ☒ | ☐ Flow cytometry |
| ☒ | ☐ MRI-based neuroimaging |

