## [Peer Review File · Nature Methods]

Optimizing Xenium In Situ data utility by quality assessment and best practice analysis workflows

Corresponding Author: Dr Sergio Marco Salas

Version 0:

Decision Letter:

15th Jan 2023

Dear Mr Marco Salas,

Thank you for your inquiry about submitting your manuscript, "Optimizing Xenium In Situ data utility by quality assessment and best practice analysis workflows" to Nature Methods. The paper sounds like it may be of interest, and should fit the scope of the journal. We would be willing to read and evaluate the full manuscript.

Of course, it is very difficult to judge a paper based only on the limited information available in a presubmission inquiry. Therefore I am sure you understand that we cannot promise to send your paper out for peer review and must read it in its entirety before deciding if this would be suitable.

Please keep in mind that the journal is aimed at a large, interdisciplinary audience and places a strong emphasis on the practical value of the work presented. We strongly encourage you to include data to validate method performance and demonstrate its general applicability.

Manuscripts describing new algorithms and software should describe the principles underlying the algorithms in an accessible style targeted for our broad biological audience. The manuscript should demonstrate the practical relevance of the tool and any advantages and limitations it presents compared to available software. The code, and ideally a user manual and example data for testing the code, must be made available to reviewers as part of the peer review process.

You will find our Guide to Authors at <http://www.nature.com/naturemethods> to assist you in preparing your manuscript. However, it is not necessary at this stage to spend major effort adhering to our detailed formatting instructions.

Thank you for your interest in Nature Methods.

Sincerely,

Lin Tang, PhD
Senior Editor
Nature Methods

Version 1:

Decision Letter:

17th May 2023

Dear Mr Marco Salas,

Your Analysis, "Optimizing Xenium In Situ data utility by quality assessment and best practice analysis workflows", has now been seen by 3 reviewers. As you will see from their comments below, although the reviewers find your work of potential interest, they have raised a number of important concerns. We are interested in the possibility of publishing your paper in Nature Methods, but would like to consider your response to these concerns before we reach a final decision on publication.

We therefore invite you to revise your manuscript to fully address all these concerns. A comprehensive benchmarking analysis of a large number of computational methods for multiplexed ISS/ISH data in general would certainly be a strong paper, but considering the current scope/topic of the paper, we will not request a complete refocus of the paper to that topic. That said, additional analysis in this aspect following Reviewer 2's suggestion is needed. Furthermore, please ensure the paper is balanced and transparent. In light of the concern raised about potential conflict of interest, we suggest Dr. Nilsson provide more detail about his advisory role for 10X Genomics.

Link Redacted

We hope to receive your revised paper within 3 months. If you cannot send it within this time, please let us know. In this event, we will still be happy to reconsider your paper at a later date so long as nothing similar has been accepted for publication at Nature Methods or published elsewhere.

OPEN SCIENCE REQUIREMENTS

REPORTING SUMMARY AND EDITORIAL POLICY CHECKLISTS

DATA AVAILABILITY

All novel DNA and RNA sequencing data, protein sequences, genetic polymorphisms, linked genotype and phenotype data,

gene expression data, macromolecular structures, and proteomics data must be deposited in a publicly accessible database, and accession codes and associated hyperlinks must be provided in the "Data Availability" section.

CODE AVAILABILITY

Please include a "Code Availability" subsection in the Online Methods which details how your custom code is made available. Only in rare cases (where code is not central to the main conclusions of the paper) is the statement "available upon request" allowed (and reasons should be specified).

For more information on our code sharing policy and requirements, please see: <https://www.nature.com/nature-research/editorial-policies/reporting-standards#availability-of-computer-code>

MATERIALS AVAILABILITY

ORCID

Sincerely,

Lin Tang, PhD
Senior Editor
Nature Methods

Reviewers' Comments:

Reviewer #1:

Remarks to the Author:
Summary of the key results

Salas et al. explore data from the new Xenium in situ platform with a focus on mouse brain and human breast cancer. Initially, Xenium datasets are compared with competing spatial transcriptomics technologies. Afterwards, key steps and the corresponding tools/algorithms for Xenium data processing are compared and an optimal workflow suggested: Identification of nuclei using Cellpose followed by read assignment using Baysor. Once cells are segmented, cell type identification was performed via filtering, log-transformation and normalization, and identification of the main PCs followed by dimensionality reduction and clustering.

Overall, the manuscript addresses very important and timely questions: How do we compare SRT methods and how do we process the individual methods robustly and efficiently. Salas et al. provide a great overview of the current state of art analysis landscape, which they comprehensively apply to “publicly” available Xenium data. The suggested best practices are a great starting point for Xenium data analysis.

Unfortunately, most described methods are not quantitatively benchmarked. This manuscript rather compares existing methods and illustrates its advantages and disadvantages. The insights and comparisons are often difficult to understand – specifically with limited prior-knowledge of the mouse brain. Additionally, the breast cancer samples seem to disappear in the follow up analysis, which illustrates the limited generalization. Finally, this manuscript does not include novel methods, data sets or biological/clinical insights, which will limit its impact.

Minor comments:

Introduction:
How do methods like Ligh-Seq (<https://doi.org/10.1038/s41592-022-01604-1>) fit into the landscape?

Results:
Where do I see FF vs FFPE comparison in Figure 1B?
“multisection” corresponds to “multi” in Figure 1B?
Please specify which methods are ISS-based in Figure 1C to enable a better comparison.
Was mesmer/deepcell segmentation tested as well?
Page 29-30: Page break

Supplement:
Figure S3 B: Show which areas were expanded. Remove comment (“Annotation would go here (Catarina)”)

Reviewer #2:

Remarks to the Author:

The investigation presented in the paper has the potential to be helpful; however, the overall content of the paper appears to be more of an advertisement and user manual for Xenium. This raises concerns about potentially misleading information for the research community, as there is a conflict of interest with the corresponding author being an advisor for 10X Genomics. It is unclear whether the main goal of the paper is to assert that Xenium wet-lab measurement is the best among all the ISS and ISH methods or to provide an objective assessment of various computational methods for multiplexed ISS/ISH data in general. If the former is the case, then the paper is not suitable for publication in Nature Methods, as the method used by Xenium has already been published. If the latter is the case, the authors should conduct a more objective evaluation by including other ISS/ISH datasets in addition to the Xenium data.

Several specific concerns have been identified:

1. The evaluation of Xenium data compared to other datasets is problematic and can be misleading. Many of the cited ISS and ISH datasets (e.g., MERFISH, STARmap) used gene panels targeting the entire mouse brain, whereas the scRNA-seq dataset used (Yao et al) for comparison only focuses on the cortex and hippocampus. To ensure a fair comparison, the authors should either use whole-brain single-cell datasets as a reference or consider the intersection of gene panels shared by all targeted mapping methods for the comparison. Additionally, the authors should clearly label the number of genes profiled in Figure 1c, in addition to the SRT/scRNA ratio.
2. The absence of the cell segmentation codes on Github makes it difficult to evaluate the comparison results.
3. The paper lacks Clustermap segmentation results in Figure 2 and Extended Data Figures 4 and 5. It is important to include these results. Furthermore, the STARmap dataset mentioned in the paper utilized Clustermap with DAPI dots for cell segmentation. Therefore, the authors should re-analyze Clustermap using DAPI sampling, which would allow for a comparison to be made with Baysor+Cellpose (DAPI).
4. The subcellular analysis presented in Figure 3F-G is unclear in the main text and figure captions. If the understanding is correct, the numbers “7” and “80” in Figure 3G represent the percentage or number of reads in the cytosol or nuclei of Astro1. However, this is inconsistent with the typical observation that nuclear RNAs only account for 10-20% of total cellular RNA.
5. The evaluation of gene imputation lacks several commonly used tools, such as Seurat, LIGER, and recently published ones

(<https://www.nature.com/articles/s41587-021-01006-2>).

6. Figure 5G, which pertains to the evaluation of spatial domain segmentation, lacks a legend, making it difficult to assess the performance of different methods. Please include a clear legend for better understanding and evaluation.

Reviewer #3:

Remarks to the Author:

Salas and colleagues present an evaluation of different analysis strategies for Xenium - a new spatial transcriptomics assay from 10x Genomics. Their evaluation examines assay detection rates, correspondence to scRNA-seq, issues related to cell segmentation, segmentation-free analysis, gene imputation, and detection of spatial domains. The manuscript represents a quick practical guidance, and not a thorough benchmarking analysis – something that the authors readily admit themselves in the Discussion. Without passing judgement on whether the current journal is the optimal choice for this manuscript type, I think such practical guidance is timely, given that Xenium assay is quite new and will likely be a popular choice for many biologists. Nevertheless, I think some steps can be taken to improve the presented evaluations and the resulting guidelines. Finally, I think everyone would benefit from more in-depth analysis of the assay characteristics.

Major points:

1. Signal to noise ratio. Fig. 1B,C and the relevant sections focus on how detection sensitivity in Xenium compares to that of other assays. This is very useful. Missing, however, is a closely analysis of how much of the detected signal represents the actual signal or non-specific background. In my experience, such signal to noise ratio can vary drastically between assays. Such estimates, though not trivial, could be obtained in a variety of ways. For example, examining intercept terms in the dependency between Xenium and scRNA-seq output.

Since the abstract does promise an evaluation of Xenium assay, it would be useful to clearly compare all key technical characteristics between assays somewhere in the manuscript.

2. Reproducibility and difference between organisms. The analysis almost entirely focuses on the mouse samples. In terms of technical characteristics, is there a difference between mouse and human samples? What is the reproducibility of the Xenium assay? Hopefully, agreement between mouse samples is notably higher than the difference between distinct human samples.

3. Segmentation benchmarking. The analysis of segmentation results is probably the most thorough in the manuscript. Demonstration of how cell types break up into region-specific clusters, and evaluation of Negative Marker Purity are both quite telling. NMP measure, however, should be sensitive expression frequency of the chosen genes. For instance, adding a gene that is virtually never detected to the panel will simply increase the measure and will not be informative. Considering realistic non-specific background rates, a “purity” measure that always scores above 95% is likely overestimated. How sensitive are the overall results to the minimum expression threshold in the positive cell type? Also, it seems the results should be limited by the non-specific detection rate. If the authors can estimate that (point 1 above), perhaps some kind of a practical expectation for optimal performance could be added.

4. The authors note negative correlation between detection efficiency and the expression magnitude in Xenium, and show that it is at least in part due to the number of probes targeting different genes. But it is unclear how much of the variation is explained by that. In other words, how well does detection of different genes in Xenium correspond to scRNA-seq detection rates after accounting for the number of probes effects? How does that compare to other protocols?

5. (not major, but related to the above) The first sentence on line 243 is unclear. If the intent is to evaluate overall agreement between scRNA-seq and Xenium, analyzing scatter plots for individual cell types (or the magnitudes predicted based on the number-of-probes model suggested in item 4 above) would be more telling. Ideally, the correspondence should be also compared between the technologies.

Minor points:

6. The abstract notes that “.. we explore eight preview Xenium datasets of the mouse brain and two of human breast cancer by comparing scalability, resolution, data quality, capacities and limitations with eight other spatially resolved transcriptomics technologies.” To be fair, even with the suggested additions, I think these claims need to be moderated, since the detailed analysis is focused on the handful of mouse datasets and comparison with other assays is quite superficial.

7. In the Introduction, the authors note that Xenium protocol produces 220 reads per cell. However, this number seems to come from the 10x default segmentation, which over-expands cell boundaries. Baysor + CellPose, which is suggested as an improved alternative yields only around 120 reads per cell (EDF 5).

8. Please expand on the ED Fig. 2A legend. What normalizations are used? How were the genes selected (there appears to be relatively few of them)

9. The calculation of the benchmark results in the provided notebook (`comparson_between_techniques.ipynb`) doesn't match those shown in the paper (e.g. `2.comparison_between_datasets/ratio_per_tech.pdf` compared to Fig. 2C)

10. line 338: “The implementation of new segmentation algorithms that should consider the 3-dimensional structure of the data” Baysor publication mentions a 3-d segmentation implementation. Would be useful if the authors commented on whether it improved the results.

11. line 347: Bento mention is a bit surprising, given that no examples are shown
12. Please comment on the computational / runtime resource requirements for different analysis
13. Please add an interpretable scale to the Fig 2D color bar
14. Please add color legend to Fig. 5G
15. Please provide code or commands for how the segmentation was run with different methods (4_segmentation_benchmark folder in github is empty). Same for sub-cellular structure analysis with Points2Regions.
16. Does nuclear-only segmentation match better to snRNA-seq and expanded segmentations to scRNA-seq?

Version 2:

Decision Letter:

25th Apr 2024

Dear Mr Marco Salas,

Your Analysis, "Optimizing Xenium In Situ data utility by quality assessment and best practice analysis workflows", has now been seen by 3 reviewers. As you will see from their comments below, our reviewers still raise a number of important concerns. We are interested in the possibility of publishing your paper in Nature Methods, but would like to consider your response to these concerns before we reach a final decision on publication.

We therefore invite you to revise your manuscript to address these concerns. Among other revisions, it is important to improve the code/data availability in the GitHub repository and make an end-to-end pipeline as suggested by Reviewers 1 and 2. We also think comparison results with other technologies are useful information although including a wider variety of tissue types is not mandatory.

Link Redacted

We hope to receive your revised paper within 3 months. If you cannot send it within this time, please let us know. In this event, we will still be happy to reconsider your paper at a later date so long as nothing similar has been accepted for publication at Nature Methods or published elsewhere.

OPEN SCIENCE REQUIREMENTS

REPORTING SUMMARY AND EDITORIAL POLICY CHECKLISTS

Please submit these with your revised manuscript. They will be available to reviewers to aid in their evaluation if the paper is

re-reviewed. If you have any questions about the checklist, please see <http://www.nature.com/authors/policies/availability.html> or contact me.

DATA AVAILABILITY

All novel DNA and RNA sequencing data, protein sequences, genetic polymorphisms, linked genotype and phenotype data, gene expression data, macromolecular structures, and proteomics data must be deposited in a publicly accessible database, and accession codes and associated hyperlinks must be provided in the "Data Availability" section.

CODE AVAILABILITY

Please include a "Code Availability" subsection in the Online Methods which details how your custom code is made available. Only in rare cases (where code is not central to the main conclusions of the paper) is the statement "available upon request" allowed (and reasons should be specified).

MATERIALS AVAILABILITY

ORCID

Sincerely,

Lin Tang, PhD
Senior Editor
Nature Methods

Reviewers' Comments:

Reviewer #1:

Remarks to the Author:

Thank you very much for submitting a revised version of the manuscript. Unfortunately, I remain critical about the scope of the manuscript. On the hand, (i) it aims to perform a comparison between the currently available experimental methods. On the other hand, (ii) it aims to provide "best practice" specifically tailored for Xenium data analysis.

(i) In my opinion, the methods comparison is now improved, but it remains difficult to generalize the results. As a user, how will these results help me choose the best experimental platform? What are the advantages and disadvantages to be aware of?
(ii) While also the best practice part is improved in comparison to the first submission, it does lack the promised [348-349] "End-to-end" pipeline: At least when following the link provided with the manuscript (https://github.com/Moldia/Xenium_benchmarking), I do not see an end-to-end pipeline, but rather an assembly of notebooks specific for this manuscript. Without documentation and more general examples this cannot be considered a pipeline helpful for the community.

In my opinion, the manuscript should focus on either "best practice" specifically tailored for Xenium data analysis - which has to include a well-documented and generalizable "End-to-end" pipeline – or an experimental methods comparison without a focus on Xenium. The later, should provide a more general overview across tissues, technologies, and downstream analysis from a user perspective. In other words: What are the advantages and disadvantages and what technology fits my question best?

Additional comment/question:

- Subcellular information in Figure F-I is described very briefly. It is mentioned that the information is important, but how important is it? Could you identify novel cell types or improve current once?

Remarks on code availability

I do not see an end-to-end pipeline, but rather an assembly of notebooks specific for this manuscript. Without documentation and more general examples this cannot be considered a pipeline helpful for the community.

Reviewer #2:

Remarks to the Author:

The manuscript has been substantially improved over the revision. But there are still a few remaining issues:

1. Line 97, "nuclear segmentation masks contain sufficient information to decipher the identity of individual cells in situ". However, given an absence of CA2-FC-IG Glut in Figures 1c,d, the statement is questionable and needs clarification. This raises doubts about the Xenium dataset's capacity to accurately define finer cell types and states, particularly in the context of highly diverse neurons. There lacks clarity regarding the level of classification of cell type and the correlation of cell types in Figure 1c,d with those in previously reported single-cell datasets and recently established spatial mouse brain cell atlases from BICCN: like eight papers described in BICCN 2.0 using MERFISH, STARmap, and Slide-seq: <https://www.nature.com/collections/fjihbeccbd>
Additionally, it is challenging to discern labels and cells in Figures 1c,d.

2. There is absence of considering 3D segmentation methods. The authors overlook available 3D nuclear segmentation methods like StarDist-3D and lack similarity comparison between 3D and 2D or segmentation-free analysis.

3. Line 153-158, 'We chose instead to focus on quantifying gene-specific characteristics of the different assays. ... Overall, we observed Xenium to be the most sensitive ISS-based technique, presenting a sensitivity similar to ISH-based technologies such as MERSCOPE and Molecular cartography (Figure 2C, Extended Data Figure 4 C-D)'. However, those comparisons are conducted among datasets with different numbers of gene panels and different brain tissue positions/regions. This inconsistency raises concerns about the accuracy and completeness of the figure.

4. Line 175-183, 'To explore the latter, we implemented a negative co-expression purity (NCP) metric. ... These results were found to be consistent when removing all highly expressed genes, which could result in lower NCP scores due to a broader expression across cells (Extended Data Figure 4E)'. However, in Extended Data Figure 4E, no negative co-expression purity (NCP) scores seem to be reported. The numbers on Extended Data Figure 4E appear to be a gene-specific efficiency ratio, which reads confusing and disorganized as gene-specific characteristics seem to be a metric discussed in the previous paragraph. Clarifying the NCP scores and relevance with gene-specific efficiency ratio would aid in understanding the figure.

5. Line 255, 'We considered omitting steps and used multiple hyperparameters within each (Online methods).' However, no detailed explanations of multiple hyperparameters can be found in the Methods part.

6. Line 305, 'Remarkably, this didn't apply to the algorithm used to identify HVFs, which managed to detect all control probes as non-variable features, while still consistently detecting ~18% of the genes as HVFs.' There is a lack of supporting figures or data for this argument. Including relevant data visualizations could substantiate the claim.

7. Line 348, the authors claim, ' We have condensed this information in an end-to-end pipeline (Code Availability) to help Xenium users to take the most out of their data.' However, it is not supported by the current code deposition provided GitHub link.

(1). The GitHub repository provided appears to be a work in progress, missing content under 'folder structure' content and a broken link to Xenium datasets (<https://www.10xgenomics.com/xenium-preview-data>). There is no documentation for clarity, making it challenging for individuals other than the authors to grasp the intended analysis workflow.

(2). Datasets used as input is not accessible or accompanied with the code, which complicates the navigation and replication of the analyses presented in the manuscript. If a dataset is large enough to be hosted on GitHub, providing some sample datasets on GitHub would facilitate a more straightforward benchmarking process for users looking to assess the code's functionality and performance.

(3) Currently, the GitHub link serves as a repository for code used in the analysis rather than an end-to-end pipeline. In addition, as an analysis resource, tutorials or vignettes are absent. Given these issues, it does not seem likely that an end-to-end pipeline will be made available shortly, as might be anticipated by users. Enhancing the repository with detailed documentation, including step-by-step tutorials and example datasets, would greatly enhance its value as a resource, facilitating a better understanding and application of the paper's findings and methodologies.

8. Assessing computational tools to explore tissue architecture is limited to the cortical region, which may not reflect the complexities of other tissues, such as the midbrain, hindbrain, or cancer tissues. An objective comparison should include a wider variety of tissue types.

In summary, the quality of many figures and benchmark analyses is below the bar of a typical NM publication, and some figures do not match the manuscript's text. The current code deposition is incomplete and difficult for potential users to navigate and replicate due to insufficient documentation and data availability.

Remarks on code availability

Due to insufficient documentation and data availability, the current code deposition is incomplete and difficult to navigate and replicate.

Reviewer #3:

Remarks to the Author:

I think that the revised manuscript has introduced substantial improvements. It offers valuable insights into the technical aspects of various experimental protocols and spatial transcriptomics analysis methods.

While I have several further suggestions, overall I find the manuscript helpful and would like to see it published. In addition to specific comments below, I also think general readability could be improved.

Major comments:

- Extended Data Figure 4B shows extremely low percentages of molecules assigned per cell (all <35%), which are very different from Figure 1B (all >60%). If I understand correctly, the reason for such a difference is differences in segmentation methods. But if that's the case, having <10% of molecules assigned to cells on MERFISH data raises questions about quality of the performed Cellpose segmentation. Again, if I understood correctly, this Cellpose segmentation was used to produce all the results in Figure 2. If so, all the important findings from Figure 2 could be compromised by these problems in segmentation.

- It would be very helpful if the authors could publish their simulated datasets with corresponding metadata, so other researchers could run the benchmarks for their methods. This way the researchers could obtain results comparable to the ones presented in the manuscript without the need to rerun all the benchmarking.

Minor comments:

- On page 5, the authors note: "Broadly speaking, the number of reads per cell scaled with the size of the panel." While some correlation can be seen in the results, potential mechanisms that could cause such a dependency are unclear. The trend seems to be largely relying on the CosMx data, which has both the highest number of measured molecules per cell and genes per panel. However, CosMx has an extremely high false-positive rate. So, in CosMx, the number of molecules per cell would increase with the number of genes, but it would be mostly driven by the false-positive molecules. I would suggest either providing support for this claim or removing it, as it can be misleading.

- It is unclear what cell segmentation was used for benchmarking all processing of spatial data. Please specify it in Methods.

- There are quite a few typos and mistakes in figure references. Some examples:

- "observing a similar number of molecules detected per cell in Xenium and other SRT commercial platforms (Figure 2E)." - Should be 2B

- "we discerned disparities in the subcellular distribution of profiled reads among various technologies (Figure 2E)" - Should be 2F
- "We found that, as a mean, transcripts beyond 10.71 μm from the cell centroid presented a higher gene expression correlation to domain-specific background signatures compared to nuclear cell type-specific signatures (Figure 3A-B, Extended Data Figure 4 A-B)" - Extended Data Figure 4A-B don't seem to be related to the claim.

Version 3:

Decision Letter:

Our ref: NMETH-AS51432C

1st Oct 2024

Dear Dr. Marco Salas,

Thank you for submitting your revised manuscript "Optimizing Xenium In Situ data utility by quality assessment and best practice analysis workflows" (NMETH-AS51432C). It has now been seen by the original referees and their comments are below. The reviewers find that the paper has improved in revision, and therefore we'll be happy in principle to publish it in Nature Methods, pending minor revisions to satisfy the referees' final requests and to comply with our editorial and formatting guidelines.

TRANSPARENT PEER REVIEW

ORCID

Sincerely,

Lin Tang, PhD
Senior Editor
Nature Methods

Reviewer #1 (Remarks to the Author):

Thank you for sharing the revised manuscript.

- The code is now improved and provides a documentation for the end-to-end analysis of Xenium data. The workflow is implemented in a notebook.

- The technology comparison focuses primarily on brain structures, which – as mentioned by the author – makes sense since "ground-truth" is available. This should be clearly stated when discussing the comparison and performance of the various platforms. The results may be very different when applied to other tissue types.

- Figure 3F: Use random order for visualization and/or improve image quality.

Reviewer #1 (Remarks on code availability):

The code is now improved and provides a documentation for the end-to-end analysis of Xenium data. The workflow is implemented in a notebook.

Reviewer #3 (Remarks to the Author):

The manuscript changes, clarifications and importantly the code released by the authors are all very helpful, and have answered my questions. This is a fast-moving field and I think this publication would be helpful and timely for the community.

Reviewer #3 (Remarks on code availability):

The repository contains notebooks for reproducing the figures and carrying out relevant analysis. It is more of a collection of recipes rather than a benchmarking suite.

Version 4:

Decision Letter:

4th Feb 2025

Dear Dr Marco Salas,

I am pleased to inform you that your Analysis, "Optimizing Xenium In Situ data utility by quality assessment and best practice analysis workflows", has now been accepted for publication in Nature Methods. The received and accepted dates will be 13th Feb 2023 and 4th Feb 2025. This note is intended to let you know what to expect from us over the next month or so, and to let you know where to address any further questions.

Over the next few weeks, your paper will be copyedited to ensure that it conforms to Nature Methods style. Once your paper is typeset, you will receive an email with a link to choose the appropriate publishing options for your paper and our Author Services team will be in touch regarding any additional information that may be required. It is extremely important that you let us know now whether you will be difficult to contact over the next month. If this is the case, we ask that you send us the contact information (email, phone and fax) of someone who will be able to check the proofs and deal with any last-minute problems.

Please feel free to contact me if you have questions about any of these points. Thank you very much for publishing your paper at Nature Methods!

Best regards,

Lin Tang, PhD
Senior Editor
Nature Methods

** Visit the Springer Nature Editorial and Publishing website at http://editorial-jobs.springernature.com?utm_source=ejP_NMeth_email&utm_medium=ejP_NMeth_email&utm_campaign=ejp_Nmeth >www.springernature.com/editorial-and-publishing-jobs for more information about our career opportunities. If you have any questions please click [here](mailto:editorial.publishing.jobs@springernature.com) . **

Open Access This Peer Review File is licensed under a Creative Commons Attribution 4.0 International License, which permits use, sharing, adaptation, distribution and reproduction in any medium or format, as long as you give appropriate credit to the original author(s) and the source, provide a link to the Creative Commons license, and indicate if changes were made. In cases where reviewers are anonymous, credit should be given to 'Anonymous Referee' and the source. The images or other third party material in this Peer Review File are included in the article's Creative Commons license, unless indicated otherwise in a credit line to the material. If material is not included in the article's Creative Commons license and your intended use is not permitted by statutory regulation or exceeds the permitted use, you will need to obtain permission directly from the copyright holder.

Dear Dr. Lin Tang,

We are pleased to resubmit a revised version of our manuscript titled “Optimizing Xenium In Situ data utility by quality assessment and best practice analysis workflows” for consideration for publication in Nature Methods as an Analysis article.

Following the actions nicely suggested by reviewers, we have in this revised version of the manuscript, (1) extended the number of Xenium datasets used throughout the study, (2) generalized the majority of our claims by expanding the analysis to newer datasets and (3) expanded and diversified the metrics used to measure the performance of different algorithms in specific tasks. We have also extended the comparison between ST commercial products, including datasets from most relevant commercial platforms. Finally, and following one of the main scopes of the manuscript, we included a new section focused on defining best practices for data preprocessing. Since several sections have been largely improved or added, we acknowledge that the claims exposed in the first version of the manuscript have been extended. Therefore, if either the editor or reviewers consider some of these sections to be expendable, we would be open to excluding them in benefit of the rest of the manuscript. All the modifications done in the manuscript are highlighted in red

Below you will find a detailed point-by-point response to the comments, concerns and suggestions raised by the reviewers.

Reviewer #1:

Remarks to the Author:

Summary of the key results

Salas et al. explore data from the new Xenium in situ platform with a focus on mouse brain and human breast cancer. Initially, Xenium datasets are compared with competing spatial transcriptomics technologies. Afterwards, key steps and the corresponding tools/algorithms for Xenium data processing are compared and an optimal workflow suggested: Identification of nuclei using Cellpose followed by read assignment using Baysor. Once cells are segmented, cell type identification was performed via filtering, log-transformation and normalization, and identification of the main PCs followed by dimensionality reduction and clustering.

Overall, the manuscript addresses very important and timely questions: How do we compare SRT methods and how do we process the individual methods robustly and efficiently. Salas et al. provide a great overview of the current state of art analysis landscape, which they comprehensively apply to “publicly” available Xenium data. The suggested best practices are a great starting point for Xenium data analysis.

Unfortunately, most described methods are not quantitatively benchmarked. This manuscript rather compares existing methods and illustrates its advantages and disadvantages. The insights and comparisons are often difficult to understand – specifically with limited prior-knowledge of the mouse brain. Additionally, the breast cancer samples seem to disappear in the

follow up analysis, which illustrates the limited generalization. Finally, this manuscript does not include novel methods, data sets or biological/clinical insights, which will limit its impact.

1. Lack of Quantitative Benchmarking: We acknowledge the concern about the absence of quantitative benchmarking through the manuscript. In this revised version of our manuscript, we have expanded and diversified the metrics used to explore the performance of each algorithm on each task described. Examples of analysis that we consider to be qualitatively benchmarked are: the comparison of segmentation algorithms, domain identification methods, SVF identification algorithms or preprocessing workflows.

2. Generalization of our claims: Due to the limited number of samples available when the first version of our manuscript was submitted, we focused our attention on the analysis of the mouse brain datasets. In this revised version, where 25 different datasets are included, we generalized some of our claims by extending our analysis to several datasets, when possible. However, we still kept the 7 mouse brain Xenium datasets as the central datasets in our study since they offered us unique properties such as (1) being a very structured tissue, (2) the availability of datasets generated with other ST platforms profiling the same tissue and (3) the availability of a good reference scRNAseq datasets to compare the datasets against.

4. Lack of Novelty: the manuscript has been submitted to Nature Method as an Analysis. This type of article is described as “ a report presenting comprehensive performance comparisons of established, related methods or tools, of key importance to a field of research” . Our manuscript, following the aim of this format, provides a comprehensive comparison of (1) both commercially available SRT platforms and technologies and (2) available algorithms used to perform some of the most important analysis typically done with this type of datasets such as segmentation, SVF identification of gene imputation. In addition, we describe for the first time some of the features of these types of assays, identifying important limitations and proposing best practices. The revised version of our manuscript further extends on these lines.

Minor comments:

Introduction:

How do methods like Ligh-Seq (<https://doi.org/10.1038/s41592-022-01604-1>) fit into the landscape?

The methodology described by Ligh-Seq would fall under a different category of ST-based methods, typically described as “Deterministic spatial barcoding technologies” in the recent review by Moffit et al. 2022. These methods typically capture the transcriptomic profile of specific ROI rather than the profile across the entire tissue. Due to space limitations, their description was left out of the first version of the manuscript. In this newest version, we have modified the introduction to briefly mention these types of technologies.

Results:

Where do I see FF vs FFPE comparison in Figure 1B?

In the previous version this information wasn't visible in panel B from Figure 1. However, in the revised version of the manuscript we have included a panel (Figure 1B) where this information is available for all datasets.

"multisection" corresponds to "multi" in Figure 1B?

This panel has been completely modified in the revised version of the manuscript and, therefore, this comment is not applicable any more.

Please specify which methods are ISS-based in Figure 1C to enable a better comparison.

The comparison between technologies has now been expanded in Figure 2. Out of all technologies available, only HS-ISS is ISS based, besides Xenium. We have modified the text to make clear the reader understands this.

Was mesmer/deepcell segmetation tested as well?

The revised version of the benchmarking of segmentation methods includes additional algorithms such as "Bayor + Xenium nuclear segmentation" or MESMER.

Page 29-30: Page break

In the revised version of the manuscript we have corrected this typo

Supplement:

Figure S3 B: Show which areas were expanded. Remove comment ("Annotation would go here (Catarina)")

Supplementary figure 3 has now been modified and the comment has been removed.

Reviewer #2:

Remarks to the Author:

The investigation presented in the paper has the potential to be helpful; however, the overall content of the paper appears to be more of an advertisement and user manual for Xenium. This raises concerns about potentially misleading information for the research community, as there is a conflict of interest with the corresponding author being an advisor for 10X Genomics. It is unclear whether the main goal of the paper is to assert that Xenium wet-lab measurement is the best among all the ISS and ISH methods or to provide an objective assessment of various computational methods for multiplexed ISS/ISH data in general. If the former is the case, then the paper is not suitable for publication in Nature Methods, as the method used by Xenium has

already been published. If the latter is the case, the authors should conduct a more objective evaluation by including other ISS/ISH datasets in addition to the Xenium data.

We appreciate the feedback and suggestions. We are aware that some of the aspects of the first version of our manuscript were unclear. We have tried to address this concerns both in the text and figures, but we'd like to clarify some of the aspects in this rebuttal letter as well:

- 1. Conflict of interests:** Mats Nilsson was a Scientific advisor for 10xGenomics until October 2023 when he left that assignment. The advisory role comprised providing strategic advice upon request from 10xGenomics. The work on this manuscript was not a part of the advisory assignment, and has never been discussed in that context. The link between "Mats Nilsson the advisor" and "Mats Nilsson the author", is that the Xenium instrument basically runs on chemistry that was developed first in Nilsson's laboratory, and then in the spin out company Cartana, that 10xGenomics acquired. Nilsson did not receive shares in 10xGenomics upon the acquisition. Having developed and applied the method pre-Xenium, Nilsson's lab generated a lot of experience interpreting this kind of data using standard epi-fluorescence microscopes and manual processing. We were thus well experienced and curious to conduct the present work together with colleagues in the field. We clarified this in the "Competing interests" section of the manuscript
- 2. Aim of the manuscript:** the aim of the manuscript is not to prove that Xenium is performing better than other ISS/ ISH methods in a set of metrics. The aim of this manuscript is to describe for the first time the datasets generated with a novel commercial product. An essential part of this analysis is to position this technology in comparison with other commercial products. In addition, we also aim to define the algorithms that perform better when applied to Xenium datasets for several common analytical tasks. All in all, our aim in this manuscript is to provide a solid and independent analysis of a new type of datasets that are becoming increasingly popular in cellular biology, serving our work as a starting point for scientists aiming to analyze this type of data. As suggested by editors and reviewers, we further expanded the datasets used through the study, the algorithms included in the benchmarking of each specific task and the number and diversity of metrics used to identify the top performer algorithms on each case.

Several specific concerns have been identified:

1. The evaluation of Xenium data compared to other datasets is problematic and can be misleading. Many of the cited ISS and ISH datasets (e.g., MERFISH, STARmap) used gene panels targeting the entire mouse brain, whereas the scRNA-seq dataset used (Yao et al) for comparison only focuses on the cortex and hippocampus. To ensure a fair comparison, the authors should either use whole-brain single-cell datasets as a reference or consider the intersection of gene panels shared by all targeted mapping methods for the comparison.

Additionally, the authors should clearly label the number of genes profiled in Figure 1c, in addition to the SRT/scRNA ratio.

To enable a fair comparison, we have used a whole mouse brain single cell dataset (ABC Atlas) as a reference. Since the new single cell dataset is coupled with a MERFISH-based spatial atlas, we took the chance to further subset the single cell dataset, including solely the cell types identified in the coronal MERFISH sections corresponding to the anteroposterior position of the profiled Xenium slides. Furthermore, since different technologies profiled different slightly different structures (i.e. cortex, hippocampus, thalamus), we annotated and subsampled each of the regions on each of the datasets, including the scRNAseq reference atlas, before comparison. As a consequence, we are computing efficiency, specificity and subcellular localization-related metrics for each of the regions independently.

In the revised manuscript, we have additionally included a panel (Figure 2B), which summarizes the number of profiled genes, overall reads/cell and genes/cell detected for each method.

2. The absence of the cell segmentation codes on Github makes it difficult to evaluate the comparison results.

Unfortunately the cell segmentation code wasn't available upon submission. In this revised version, all the code in the main github repository has been updated and reorganized to facilitate the reproducibility and use of the code presented, including the cell segmentation code.

3. The paper lacks Clustermap segmentation results in Figure 2 and Extended Data Figures 4 and 5. It is important to include these results. Furthermore, the STARmap dataset mentioned in the paper utilized Clustermap with DAPI dots for cell segmentation. Therefore, the authors should re-analyze Clustermap using DAPI sampling, which would allow for a comparison to be made with Baysor+Cellpose (DAPI).

In the revised version of the manuscript we have included clustermap when benchmarking different segmentation strategies. Furthermore, in the revised comparison between ST platforms, we have resegmented all datasets using a common segmentation method (Cellpose) to ensure a fair comparison. Due to the availability of new datasets generated using novel ST platforms and the lack of nuclear staining or position of individual reads in some of the datasets used in the initially submitted manuscript, in this revised version we have modified the datasets included in the comparison. In the revised version of the manuscript, we have included datasets generated with commercially available automated platforms (CosMx, Xenium, Molecular Cartography, Vizgen) and other ST technologies (HS-ISS, MERFISH).

4. The subcellular analysis presented in Figure 3F-G is unclear in the main text and figure captions. If the understanding is correct, the numbers "7" and "80" in Figure 3G represent the percentage or number of reads in the cytosol or nuclei of Astro1. However, this is inconsistent with the typical observation that nuclear RNAs only account for 10-20% of total cellular RNA.

The numbers “7” and “80” originally corresponded to the specific names of the subcellular clusters identified in Astrocytes. The subcellular analysis performed in our datasets, which has been reimplemented and reorganized in the new version of the manuscript pursues two main objectives: (1) confirming the detection of RNA transcripts *in situ* with a different subcellular localization, mainly confirmed by the nuclear/cytosolic differences and (2) highlighting how this differences can have an impact in the identification of cellular and subcellular structures. With this second aim and using the Points2Regions algorithm, we defined and clustered subcellular structures of the cells, demonstrating subtle but consistent differences between different cellular components. In this newer version of the manuscript, this analysis is illustrated in Figure 1 [N] . We have briefly annotated the clusters based on (1) the subcellular localization of individual clusters and (2) the cell type of origin where they were detected, as defined by the segmentation-based analysis of tissues.

5. The evaluation of gene imputation lacks several commonly used tools, such as Seurat, LIGER, and recently published ones (<https://www.nature.com/articles/s41587-021-01006-2>).

Following the advice, we have now included both Seurat and LIGER in the revised version of the manuscript. This is illustrated, for instance, in Figure 5A,B.

6. Figure 5G, which pertains to the evaluation of spatial domain segmentation, lacks a legend, making it difficult to assess the performance of different methods. Please include a clear legend for better understanding and evaluation.

This panel has been removed from the updated version of the manuscript.

Reviewer #3:

Remarks to the Author:

Salas and colleagues present an evaluation of different analysis strategies for Xenium - a new spatial transcriptomics assay from 10x Genomics. Their evaluation examines assay detection rates, correspondence to scRNA-seq, issues related to cell segmentation, segmentation-free analysis, gene imputation, and detection of spatial domains. The manuscript represents a quick practical guidance, and not a thorough benchmarking analysis – something that the authors readily admit themselves in the Discussion. Without passing judgement on whether the current journal is the optimal choice for this manuscript type, I think such practical guidance is timely, given that Xenium assay is quite new and will likely be a popular choice for many biologists. Nevertheless, I think some steps can be taken to improve the presented evaluations and the resulting guidelines. Finally, I think everyone would benefit from more in-depth analysis of the assay characteristics.

Major points:

1. Signal to noise ratio. Fig. 1B,C and the relevant sections focus on how detection sensitivity in Xenium compares to that of other assays. This is very useful. Missing, however, is a closely analysis of how much of the detected signal represents the actual signal or non-specific background. In my experience, such signal to noise ratio can vary drastically between assays.

Such estimates, though not trivial, could be obtained in a variety of ways. For example, examining intercept terms in the dependency between Xenium and scRNA-seq output.

Since the abstract does promise an evaluation of Xenium assay, it would be useful to clearly compare all key technical characteristics between assays somewhere in the manuscript.

We appreciate the constructive criticism of our previous analysis and indeed, we believe that our manuscript benefited from a further comparison between SRT technologies. In this revised version of the manuscript, we have further quantified the detection efficiency of each technology in several manners including (1) the calculation of ST/SC ratio for each gene profiled on each technology (Figure 2C), (2) a side-by-side comparison between each pair of technologies in terms of total detected reads/cell (Extended Data Figure 4E) and (3) a direct comparison of the expression of common markers in specific populations, detected across platforms (Figure 2E). We believe the use of different metrics allow us to get more robust conclusions about the detection efficiency of each technology. Together with the detection efficiency, we further explored other aspects of these technologies such as the number of profiled transcripts, number of molecules detected per cell, their detection specificity and the transcript-specific subcellular localization. In all those cases, the analysis is limited by region, ensuring the comparability of the different datasets.

2. Reproducibility and difference between organisms. The analysis almost entirely focuses on the mouse samples. In terms of technical characteristics, is there a difference between mouse and human samples? What is the reproducibility of the Xenium assay? Hopefully, agreement between mouse samples is notably higher than the difference between distinct human samples.

In the revised version of this manuscript, we have increased the number of datasets used to describe the main characteristics of data produced with Xenium, including datasets generated using samples from different species including mouse and human. By comparing both datasets we identified an overall higher detection efficiency in the mouse datasets. However, since all human samples were FFPE sections and all mouse datasets were obtained from Fresh Frozen sections, it is not possible to conclude anything regarding differences in efficiency between certain species profiled.

Regarding the reproducibility of the assay, the newest version of the manuscript includes mouse brain datasets from two different specimens and generated in two different Xenium machines. Despite these differences, both datasets presented minor differences in terms of relative gene expression and populations identified, integrating almost perfectly without any harmonization of the data, as illustrated in Extended Data Figure 1. The similarity between these datasets confirms the reproducibility of the Xenium assay, indicating that any differences observed between different tissues profiled with the same panel can be attributed to biological differences.

3. Segmentation benchmarking. The analysis of segmentation results is probably the most thorough in the manuscript. Demonstration of how cell types break up into region-specific clusters, and evaluation of Negative Marker Purity are both quite telling. NMP measure, however, should be sensitive expression frequency of the chosen genes. For instance, adding a gene that is virtually never detected to the panel will simply increase the measure and will not

be informative. Considering realistic non-specific background rates, a “purity” measure that always scores above 95% is likely overestimated. How sensitive are the overall results to the minimum expression threshold in the positive cell type? Also, it seems the results should be limited by the non-specific detection rate. If the authors can estimate that (point 1 above), perhaps some kind of a practical expectation for optimal performance could be added.

In response to the concerns about the Negative Marker Purity (NMP) measure, we extended the method section with a more detailed description of the NMP that clarifies how we generalize over different gene panels and how genes are weighted. We normalize in such a way that genes are weighted equally and not by their total number of reads, and that the NMP measure is agnostic to cell type proportions.

We agree that genes that are only sparsely detected in the spatial dataset will affect the measure (most likely the purity decreases, as falsely positive detected spots take up a larger portion of the reads for that gene). While this affects the comparability of the measure between datasets, the effect within a dataset is systematic. Therefore the purpose of using the purity measure for segmentation comparisons within datasets is not affected.

We agree that a score above 95% seems overestimated, this is due to the formulation of our metric including an unrealistic 0%-baseline that refers to the scenario where all reads are found in negative cell types only. To have a more meaningful purity measure we now scaled the metric to a baseline of random permutations of cell type labels. This way we find purity measures in the ranges of 55 - 80% (brain) and 20 - 40% (breast cancer).

Regarding defining an optimal purity based on non-specific detection rates, we calculated the optimal performance based on Xenium's "Negative control probe rate". Specifically, we simulated an optimal segmentation by removing all reads of negative markers in the respective cell types except for a read count referring to the non-specific detection rate. The negative marker purity was calculated on this theoretical read distribution. We found that this effect is smaller than our previous correction term of the optimal performance: The NMP is formulated as

$$NMP = \begin{cases} 1 - (\bar{X}_{neg}^{(sp)} - \bar{X}_{neg}^{(sc)}) & \bar{X}_{neg}^{(sp)} > \bar{X}_{neg}^{(sc)} \\ 1 & otherwise \end{cases}$$

where $\bar{X}_{neg}^{(sp)}$ refers to the reading ratio of negative markers in "wrong" (i.e. negative) cell types.

The scRNAseq data is also not "fully pure" ($\bar{X}_{neg}^{(sc)}$) since we define negative markers with some threshold based on the scRNAseq counts. Therefore we clip the NMP measure at the pureness of the scRNAseq reference. In the given data sets the non-specific detection rates effect is smaller than the scRNAseq correction. Therefore the optimal NMP does not change.

4. The authors note negative correlation between detection efficiency and the expression

magnitude in Xenium, and show that it is at least in part due to the number of probes targeting different genes. But it is unclear how much of the variation is explained by that. In other words, how well does detection of different genes in Xenium correspond to scRNA-seq detection rates after accounting for the number of probes effects? How does that compare to other protocols?

In this novel version of the manuscript, we have both modified (1) the Xenium datasets used, (2) the single cell RNA-seq of reference and (3) the regions of the brain used in the analysis. With this new set up, we could further investigate the relationship between the detection in Xenium and in scRNA-seq for the genes profiled (Extended Data Figure 4E). We identified a lack of correlation between the detection levels in both technologies for profiled genes. Briefly, lowly detected genes in scRNA-seq were detected at higher levels *in situ*, while highly detected genes in scRNA-seq were detected at comparable levels using Xenium.

As presented in the initially submitted manuscript, we hypothesized whether the number of probes used to target each gene explains this. However, after exploring the relation between the number of probes used to target each gene and the detection efficiency, we found a correlation close to 0 ($r=-0.07$). Since the relationship between the number of probes and Xenium's detection efficiency could not be generalized, we have excluded this claim from the manuscript.

5. (not major, but related to the above) The first sentence on line 243 is unclear. If the intent is to evaluate overall agreement between scRNA-seq and Xenium, analyzing scatter plots for individual cell types (or the magnitudes predicted based on the number-of-probes model suggested in item 4 above) would be more telling. Ideally, the correspondence should be also compared between the technologies.

In the revised version of the manuscript we have extended our comparison between technologies, as illustrated in Figure 2 and Extended Data Figure 4. The newest version of the manuscript compares the efficiency and specificity between platforms.

Minor points:

6. The abstract notes that “.. we explore eight preview Xenium datasets of the mouse brain and two of human breast cancer by comparing scalability, resolution, data quality, capacities and limitations with eight other spatially resolved transcriptomics technologies.” To be fair, even with the suggested additions, I think these claims need to be moderated, since the detailed analysis is focused on the handful of mouse datasets and comparison with other assays is quite superficial.

In the updated version of the manuscript we have used the different datasets through the datasets in several sections. Therefore, we believe our abstract accurately reflects the content of our manuscript.

7. In the Introduction, the authors note that Xenium protocol produces 220 reads per cell. However, this number seems to come from the 10x default segmentation, which over-expands

cell boundaries. Baysor + CellPose, which is suggested as an improved alternative yields only around 120 reads per cell (EDF 5).

As shown through our manuscript, especially in Figure 1-3, the segmentation used can largely influence the number of reads and genes detected per cell. As a consequence, we agree in the fact that it is worth clarifying the segmentation strategy used when describing the number of reads/cell and genes/cell found in different datasets. We have rewritten this section accordingly.

8. Please expand on the ED Fig. 2A legend. What normalizations are used? How were the genes selected (there appears to be relatively few of them)

Figure 2A (Figure 3A in the revised version) includes two subpanels. A first panel (left) representing a Region of Interest (ROI) of a mouse brain tissue with reads plotted on top of a background DAPI staining and colored based on their distance to their closer centroid. On the second panel (right), an example of the Pearson's correlation coefficient (PCC) between expression signature of the reads located at a specific distance to the centroid and the nuclear signature and background signature of the cell type represented in a specific cell domain is shown.

The signatures defined include all genes profiled in the panel and, thus, the PCCs computed between signatures consider the expression of all genes. The expression signatures have been normalized by the total amount of reads detected on each distance-specific signature. All these details have been further clarified in the methods section. If further details are needed, the code used to do this analysis is freely available and can be used to explore the details of the analysis. We have composed a new version of this panel for the revised version of the manuscript and modified the legend and figure caption accordingly.

9. The calculation of the benchmark results in the provided notebook (comparison_between_techniques.ipynb) doesn't match those shown in the paper (e.g. 2.comparison_between_datasets/ratio_per_tech.pdf compared to Fig. 2C)

Apologize for the inconsistencies in our available code. The revised code includes a notebook that can be used to generate the results presented in the manuscript.

10. line 338: "The implementation of new segmentation algorithms that should consider the 3-dimensional structure of the data" Baysor publication mentions a 3-d segmentation implementation. Would be useful if the authors commented on whether it improved the results.

We now conducted experiments using Baysor with and without z-coordinate information. Interestingly, incorporating additional 3D information does not enhance the precision of assigning spot clusters to cells in terms of purity. Rather, it results in an effect similar to cellular expansion, characterized by a higher proportion of assigned reads and a reduced purity of negative markers. This approach appeared advantageous for the breast cancer dataset, but for

brain data, the 2D results were more favorable due to a significant drop in purity with 3D data. Consequently, the integration of z-coordinate information should be considered a hyperparameter, subject to evaluation for each specific dataset.

11. line 347: Bento mention is a bit surprising, given that no examples are shown

Our aim including this citation was to highlight that other types of analysis are possible with Xenium datasets. In this revised version of the manuscript we have removed this sentence.

12. Please comment on the computational / runtime resource requirements for different analysis

We have included information about the runtime/resource requirements for SVF finder algorithms in Extended Data Figure 7A.

13. Please add an interpretable scale to the Fig 2D color bar

We have now modified the panel and corrected the color bar.

14. Please add color legend to Fig. 5G

Thanks for reporting the typo. We have now modified the panel and included a figure legend.

15. Please provide code or commands for how the segmentation was run with different methods (4_segmentation_benchmark folder in github is empty). Same for sub-cellular structure analysis with Points2Regions.

For the submission of the revised version of our manuscript, we have updated all the code available in our github repository.

16. Does nuclear-only segmentation match better to snRNA-seq and expanded segmentations to scRNA-seq?

This is of course a very interesting point to address. However, we believe it's out of the scope of this manuscript.

Dear Dr. Lin Tang,

We are pleased to resubmit a revised version of our manuscript titled “Optimizing Xenium In Situ data utility by quality assessment and best practice analysis workflows” for consideration for publication in Nature Methods as an Analysis article.

Following the actions suggested by reviewers, we have (1) worked on the availability of the data and code used in the study, to facilitate reproducibility of the analysis included in the manuscript, (2) developed an end-to-end pipeline for the analysis of Xenium data following our best practice guidelines, and (3) addressed the concerns raised by reviewers, revising the content of the manuscript when necessary.

Dr. Nima Rafati has contributed to the development of the end-to-end pipeline, and we have thus added him to the list of authors.

Below you will find a detailed point-by-point response to the comments, concerns and suggestions raised by the reviewers. Responses are colored in red.

Reviewer #1:

Remarks to the Author:

Thank you very much for submitting a revised version of the manuscript. Unfortunately, I remain critical about the scope of the manuscript. On the hand, (i) it aims to perform a comparison between the currently available experimental methods. On the other hand, (ii) it aims to provide “best practice” specifically tailored for Xenium data analysis.

(i) In my opinion, the methods comparison is now improved, but it remains difficult to generalize the results. As a user, how will these results help me choose the best experimental platform? What are the advantages and disadvantages to be aware of?

Our intention with our platform’s comparison is not to identify an optimal platform for all cases, but rather to expose the main benefits and limitations of different technologies in terms of panel size, detection efficiency and specificity. In those regards, we believe several insights emerge from our analysis:

- Commercial platforms (Xenium, CosMx, MERSCOPE...), unlike their home-made counterparts (ISS, MERFISH, SeqFISH...) present similar characteristics, including a high detection efficiency and large (>200 genes) panel sizes.
- CosMx currently presents the largest panel sizes and therefore, the largest number of reads/cell. However, it is also the most unspecific platform.
- Common segmentation is key to enable a fair comparison between platforms.

We have now rephrased some parts of the text to clarify the main conclusions of our analysis.

(ii) While also the best practice part is improved in comparison to the first submission, it does lack the promised [348-349] “End-to-end” pipeline: At least when following the link provided with the manuscript (https://github.com/Moldia/Xenium_benchmarking), I do not see an end-to-end pipeline, but rather an assembly of notebooks specific for this manuscript. Without documentation and more general examples this cannot be considered a pipeline helpful for the community.

In the revised version of our manuscript, we have focused on further developing the proposed end-to-end pipeline to ensure the community can make use of it, providing a general example of use. All the code and documentation can be found in the provided Github repository (https://github.com/Moldia/Xenium_benchmarking).

In my opinion, the manuscript should focus on either “best practice” specifically tailored for Xenium data analysis - which has to include a well-documented and generalizable “End-to-end” pipeline – or an experimental methods comparison without a focus on Xenium. The later, should provide a more general overview across tissues, technologies, and downstream analysis from a user perspective. In other words: What are the advantages and disadvantages and what technology fits my question best?

We believe our manuscript addresses a general curiosity in the spatial omics community, which we could summarize in the following question: “What is this novel technology and how do I get the most out of it?” In our manuscript is focused on answering this question. The first part of our manuscript focuses on characterizing the performance of Xenium and its main characteristics. In these regards, understanding how this technology performs in comparison to other available platforms is essential for truly understanding the relevance of the platform. In the second part of our manuscript, we focus on understanding how to get the most out of Xenium with the available tools, benchmarking existing algorithms in multiple tasks when applied to Xenium datasets including: cell segmentation, domain identification, spatially variable feature identification and

gene imputation. We believe these two parts are intrinsically related, and we consider it an added value that the second part can be applied to other imaging based SRT methods as well. Examples of these are the influence of the segmentation strategy employed when evaluating different platforms, or the relation of the imputation performance and the profiled genes. All in all, we believe our manuscript can serve as a reference for both people adopting these new technologies and experienced researchers who aim to get the most out of their data.

Additional comment/question:

- Subcellular information in Figure F-I is described very briefly. It is mentioned that the information is important, but how important is it? Could you identify novel cell types or improve current ones?

The differences reported in terms of subcellular distribution in our work are related to a different transcriptomic composition between nuclei and cytoplasm in the profiled cells. First of all, such differences have been shown to be biologically meaningful. For instance, Xia et al.(1) have shown how these differences can be used to characterize cellular dynamics. In addition, the ability to capture the localization of RNAs within a cell can be further used to understand RNA localization patterns within the cell, which are tightly linked to their function/ the function of their resulting protein(2).

Besides its intrinsic relevance, subcellular RNA heterogeneity is a relevant aspect of these datasets that needs to be considered in many of the most common analyses performed. As suggested by the reviewer, depending on whether only cytoplasmic reads are considered or not in the analysis, the expression signature of the cell can be slightly altered, resulting in different expression profiles. In this sense, we see subcellular RNA heterogeneity as an aspect to consider in tasks such as cell segmentation, cell type identification or cellular communication. We have now expanded our description of the importance of RNA subcellular localization in the revised manuscript.

Remarks on code availability

I do not see an end-to-end pipeline, but rather an assembly of notebooks specific for this manuscript. Without documentation and more general examples this cannot be considered a pipeline helpful for the community.

We understand the criticism regarding the lack of a well documented end-to-end pipeline in the previous version of our manuscript. We have restructured and put emphasis on generating an

end-to-end pipeline that can be run by any new user providing additional examples on how the code can be used and further documenting all the code used through this study. Please find the revised version of the code in the link (https://github.com/Moldia/Xenium_benchmarking)

Reviewer #2:

Remarks to the Author:

The manuscript has been substantially improved over the revision. But there are still a few remaining issues:

1. Line 97, "nuclear segmentation masks contain sufficient information to decipher the identity of individual cells *in situ*". However, given an absence of CA2-FC-IG Glut in Figures 1c,d, the statement is questionable and needs clarification. This raises doubts about the Xenium dataset's capacity to accurately define finer cell types and states, particularly in the context of highly diverse neurons. There lacks clarity regarding the level of classification of cell type and the correlation of cell types in Figure 1c,d with those in previously reported single-cell datasets and recently established spatial mouse brain cell atlases from BICCN: like eight papers described in BICCN 2.0 using MERFISH, STARmap, and Slide-seq: <https://www.nature.com/collections/fqihbeccbd>

With our statement we don't aim to claim that all fine-grain cell subtypes are identifiable with Xenium in a single experiment. Cell type identification *in situ* depends on a large number of factors such as the number of cells profiled and the gene panel used or the RNA quality of the tissue profiled. With such a limited dataset, it's understandable that we can't describe the cellular diversity identified in the BICCN 2.0 papers. Instead, in our sentence we intended to highlight that, by using solely nuclear segmentation masks one can identify the same cellular diversity as the one obtained when segmenting cells with the best cellular segmentation strategies (in our case, Baysor). In other words, nuclear transcripts are sufficient to identify the same cell populations detectable by processing the output of more sophisticated segmentation algorithms. This agrees observation agrees with the single cell methods, where it has been shown that scRNA-seq and snRNA-seq can generally capture the same cellular diversity(3).

To support this argument, in the revised version of the manuscript we have jointly processed the output of the best segmentation strategy identified in Figure 3 (Baysor) and the cells defined by Xenium's segmentation (nuclear reads only). We summarized these results in Figure 3F-H.

Essentially, we confirm that all populations captured by Baysor are also detectable in a comparable frequency when using only nuclear reads. Mild differences in cell type proportions are found between the two datasets, despite Baysor presenting an overall larger number of counts per cell due to improved segmentation. In addition, we have also modified the text to guarantee that our message is clearly stated.

Additionally, it is challenging to discern labels and cells in Figures 1c,d.

In this revised version of the manuscript we have modified panel 1c increasing the label sizes and facilitating its readability.

2. There is absence of considering 3D segmentation methods. The authors overlook available 3D nuclear segmentation methods like StarDist-3D and lack similarity comparison between 3D and 2D or segmentation-free analysis.

Upon submission, Xenium's output did not provide any 3D image stacks of the nuclei detected and, as a consequence, those methods could not be added to the comparison. In contrast, since three-dimensional position of the decoded reads was available, segmentation-free approaches (SSAM, points2regions) which employ the 3D coordinates of the decoded spots could be included in our analysis, as illustrated in Figure 1. In our study, we showed the capacity of multiple segmentation-free algorithms to identify the majority of the cell type-specific expression signatures while incorporating additional dimension into the analysis by considering the 3D coordinates of reads and the subcellular RNA localization patterns (Extended Data 2 and 3). Since segmentation-free algorithms do not segment individual cells, the comparison between the outputs obtained from segmentation-based and segmentation-free algorithms is challenging using the metrics used through this study. We acknowledge this comparison is definitely relevant and would be worth exploring it in future work.

3. Line 153-158, 'We chose instead to focus on quantifying gene-specific characteristics of the different assays. ... Overall, we observed Xenium to be the most sensitive ISS-based technique, presenting a sensitivity similar to ISH-based technologies such as MERSCOPE and Molecular cartography (Figure 2C, Extended Data Figure 4 C-D)'. However, those comparisons are conducted among datasets with different numbers of gene panels and different brain tissue positions/regions. This inconsistency raises concerns about the accuracy and completeness of the figure.

Since different datasets profile slightly different brain structures, it is essential for a fair comparison to include only common structures. As described in Figure 2A, we manually segmented cortical, hippocampal and thalamic regions of each dataset and only included commonly profiled regions in the comparison, as explained in the manuscript.

Since the technologies compared are targeted methods, only a subset of the expressed genes are profiled by each technology. Thus, there are very few genes commonly profiled across technologies, which complicates a systematic benchmarking of the different technologies. In our study, we propose a workaround for this issue by comparing, for each gene, its levels of expression *in situ* to the levels of expression detected in a reference single cell RNA-sequencing data obtained from comparable tissue regions. Using this logic, we don't consider whether an individual gene is highly or lowly expressed, but which is its relative expression in comparison to our reference single cell RNA-seq dataset (Figure 2) . As a consequence, our analysis aims to decipher general characteristics of each platform in terms of detection efficiency and specificity, regardless of which genes are profiled.

4. Line 175-183, 'To explore the latter, we implemented a negative co-expression purity (NCP) metric. ... These results were found to be consistent when removing all highly expressed genes, which could result in lower NCP scores due to a broader expression across cells (Extended Data Figure 4E).' However, in Extended Data Figure 4E, no negative co-expression purity (NCP) scores seem to be reported. The numbers on Extended Data Figure 4E appear to be a gene-specific efficiency ratio, which reads confusing and disorganized as gene-specific characteristics seem to be a metric discussed in the previous paragraph. Clarifying the NCP scores and relevance with gene-specific efficiency ratio would aid in understanding the figure.

We apologize for the confusion-Extended Data Figure 4E was wrongly referenced in the previous version of the manuscript. Instead, the section should refer to Extended Data Figure 4D, which corresponds to the negative co-expression purity (NCP) scores reported by each technology when excluding high expressors. We have now corrected this in the manuscript.

Regarding the relation between gene efficiency and NCP scores, in the manuscript we hypothesize that genes presenting a high efficiency (ratio>1.2) could have a lower NCP score due the detection of lowly expressed transcripts in cells where these transcripts could not be captured by scRNA-seq due to a limited detection efficiency. To exclude that this effect would affect our interpretation of the results, in the analysis presented in Supplementary Figure 4D, we excluded

the genes with an efficiency ratio above 1.2 from the calculation of NCP scores. Overall, we didn't observe major differences with the analysis done with all genes included (Figure 2D), which suggests the NCP scores are not strongly affected by the effect we described and thus, they accurately quantify the specificity of each technology.

5. Line 255, 'We considered omitting steps and used multiple hyperparameters within each (Online methods).' However, no detailed explanations of multiple hyperparameters can be found in the Methods part.

The previous version of the manuscript included Extended Data Figure 6, which presented further details on what steps were skipped in each workflow and which hyperparameters were tuned. However, this figure was only referenced in the Methods section and we understand it was difficult to link to the main text. In this revised version, we have referenced Extended Data Figure 6 in the main text. Furthermore, we have added further description about the different preprocessing workflows explored in the Methods section.

6. Line 305, 'Remarkably, this didn't apply to the algorithm used to identify HVFs, which managed to detect all control probes as non-variable features, while still consistently detecting ~18% of the genes as HVFs.' There is a lack of supporting figures or data for this argument. Including relevant data visualizations could substantiate the claim.

We acknowledge that a reference to a supporting figure was missing in the highlighted sentence. This claim is supported by the analysis presented in Figure 4 I-J, which have added to the sentence in the revised manuscript.

7. Line 348, the authors claim, ' We have condensed this information in an end-to-end pipeline (Code Availability) to help Xenium users to take the most out of their data.' However, it is not supported by the current code deposition provided GitHub link.

We have now revised code available in the provided Github repository and substantially improved it, providing an end-to-end pipeline based on the best-practices identified through our study. Additionally, we provide an example dataset and in-depth documentation of the package.

(1). The GitHub repository provided appears to be a work in progress, missing content under 'folder structure' content and a broken link to Xenium datasets (<https://www.10xgenomics.com/xenium-preview-data>). There is no documentation for clarity,

making it challenging for individuals other than the authors to grasp the intended analysis workflow.

In this revised version of the manuscript we have substantially revised the code provided and documented all functions developed for this study to guarantee the reproducibility of our findings.

(2). Datasets used as input are not accessible or accompanied with the code, which complicates the navigation and replication of the analyses presented in the manuscript. If a dataset is large enough to be hosted on GitHub, providing some sample datasets on GitHub would facilitate a more straightforward benchmarking process for users looking to assess the code's functionality and performance.

The datasets used in this study are now provided in several formats and can be accessed from the links provided both in the manuscript and in the corresponding Github repository.

(3) Currently, the GitHub link serves as a repository for code used in the analysis rather than an end-to-end pipeline. In addition, as an analysis resource, tutorials or vignettes are absent. Given these issues, it does not seem likely that an end-to-end pipeline will be made available shortly, as might be anticipated by users. Enhancing the repository with detailed documentation, including step-by-step tutorials and example datasets, would greatly enhance its value as a resource, facilitating a better understanding and application of the paper's findings and methodologies.

We have modified the code available in https://github.com/Moldia/Xenium_benchmarking according to the suggestion and it now includes tutorials, an end-to-end pipeline as well as documentation.

8. Assessing computational tools to explore tissue architecture is limited to the cortical region, which may not reflect the complexities of other tissues, such as the midbrain, hindbrain, or cancer tissues. An objective comparison should include a wider variety of tissue types.

While defining tissue structures in the mouse brain can be relatively straightforward, identifying a ground truth for other tissues, such as cancer datasets, can be complex. Since we recognise the value of extending the analysis to other structures, we decided instead to focus on different pre-annotated brain regions across the coronal sections and to compare the performance of different algorithms in this task. In this revised version of the manuscript we extended the analysis previously presented to a second brain region, which covers thalamic and hypothalamic structures

(Extended Data Figure 7B-C). The results presented are consistent with the previously reported analysis in the cortex, suggesting that our results are not region-specific.

In summary, the quality of many figures and benchmark analyses is below the bar of a typical NM publication, and some figures do not match the manuscript's text. The current code deposition is incomplete and difficult for potential users to navigate and replicate due to insufficient documentation and data availability.

In the revised version of this manuscript we have addressed the raised concerns, improving the quality of the description and available code, ensuring the reproducibility of our analysis.

Remarks on code availability

Due to insufficient documentation and data availability, the current code deposition is incomplete and difficult to navigate and replicate.

The code available in https://github.com/Moldia/Xenium_benchmarking has been updated and includes all requested information, including an end-to-end pipeline, documentation of all the functionalities as well as links to all datasets used in the study and example datasets for the end-to-end pipeline.

Reviewer #3:

Remarks to the Author:

I think that the revised manuscript has introduced substantial improvements. It offers valuable insights into the technical aspects of various experimental protocols and spatial transcriptomics analysis methods.

While I have several further suggestions, overall I find the manuscript helpful and would like to see it published. In addition to specific comments below, I also think general readability could be improved.

Major comments:

- Extended Data Figure 4B shows extremely low percentages of molecules assigned per cell (all <35%), which are very different from Figure 1B (all >60%). If I understand correctly, the reason for such a difference is differences in segmentation methods. But if that's the case, having <10% of molecules assigned to cells on MERFISH data raises questions about quality of the performed Cellpose segmentation. Again, if I understood correctly, this Cellpose segmentation was used to produce all the results in Figure 2. If so, all the important findings from Figure 2 could be compromised by these problems in segmentation.

We agree with this comment and indeed we believe that results in Figure 2 are influenced by the quality of the segmentation. This can be further seen in Figure 3, where we describe the differences in percentage of molecules assigned to cells by employing different segmentation techniques on Xenium datasets, with proportion of assigned reads ranging from 0.3 to 1 depending on the strategy used. Following the reviewer's concern, we revised the quality of the segmentation previously performed and, while overall the results were reasonably good for most of the technologies, the quality of MERFISH's segmentation was especially poor. To address this, we have resegmented all datasets using Cellpose cytoplasm model, instead of the nuclear model, which was used in the previous version of the manuscript. As a consequence of this modification, in this manuscript we present a modified version of the analysis presented in Figure 2. Although most of the findings are consistent, we for example observe important differences in MERFISH's performance, with a higher proportion of assigned reads (~50%). In this version of the manuscript we have modified the text accordingly. Furthermore, to visually illustrate the quality of the segmentation performed for each of the platforms, we have incorporated Extended Data Figure 4G. We would like to thank the reviewer for highlighting the potential poor segmentation of MERFISH and, thus, giving us the opportunity to build a more robust analysis.

- It would be very helpful if the authors could publish their simulated datasets with corresponding metadata, so other researchers could run the benchmarks for their methods. This way the researchers could obtain results comparable to the ones presented in the manuscript without the need to rerun all the benchmarking.

Simulated datasets are, essentially, an alteration of scRNA-seq datasets downloaded from CellXGene Census with the aim of mimicking spatial datasets. The raw data used as a source for our simulations can be downloaded from CellXGene Census as described in the code provided (*6_1_extract_scRNAseq_from_Census_cellxgene.ipynb*) and transformed into spatial-like

datasets (6_2_extracting_characteristics_simulated_datasets.ipynb) in a reproducible manner. Thus, we would like to propose the reviewer to use these notebooks as a data source.

Minor comments:

- On page 5, the authors note: "Broadly speaking, the number of reads per cell scaled with the size of the panel." While some correlation can be seen in the results, potential mechanisms that could cause such a dependency are unclear. The trend seems to be largely relying on the CosMx data, which has both the highest number of measured molecules per cell and genes per panel. However, CosMx has an extremely high false-positive rate. So, in CosMx, the number of molecules per cell would increase with the number of genes, but it would be mostly driven by the false-positive molecules. I would suggest either providing support for this claim or removing it, as it can be misleading.

We appreciate the reviewer's point and we acknowledge that the point we wanted to make in the text was not clear enough in our submitted manuscript. We don't aim to claim that there is any molecular mechanism that causes a dependency between the number of reads per cell and the size of the panel. Instead, we just wanted to briefly mention a very naive aspect of these assays which is that the more genes you profile in an assay, the more reads you detect. In other words, an easy way of increasing the number of reads captured per cell is to profile more genes, independently from the specificity of the assay used. This statement serves as a motivation for further investigating gene-specific detection efficiency and specificity across assays, which we believe it's a fairer comparison between platforms. In the revised version of the manuscript we have clarified this part of the text.

- It is unclear what cell segmentation was used for benchmarking all processing of spatial data. Please specify it in Methods.

In the exploration and benchmarking of different preprocessing strategies, two different types of datasets were used, as explained in the main text. First, since simulated data, consisting of spatial-like scRNA-seq datasets, was simulated based on existing scRNA-seq datasets, as described in the methods section of our manuscript, no segmentation was required for this task. In addition, preprocessing strategies were further explored using real Xenium datasets, exploring the influence of each step and hyperparameter in the final output. In this case, the output of default Xenium segmentation was used as an input, considering only nuclear reads to minimize the effect

of missegmentation in the analysis. We have now clarified this in the Methods section of the revised manuscript.

- There are quite a few typos and mistakes in figure references. Some examples:
 - "observing a similar number of molecules detected per cell in Xenium and other SRT commercial platforms (Figure 2E)." - Should be 2B
 - "we discerned disparities in the subcellular distribution of profiled reads among various technologies (Figure 2E)" - Should be 2F
 - "We found that, as a mean, transcripts beyond 10.71 μm from the cell centroid presented a higher gene expression correlation to domain-specific background signatures compared to nuclear cell type-specific signatures (Figure 3A-B, Extended Data Figure 4 A-B)" - Extended Data Figure 4A-B don't seem to be related to the claim.

Thanks for reporting the typos. We have carefully gone through all figures and corrected them accordingly.

References

1. Xia C, Fan J, Emanuel G, Hao J, Zhuang X. Spatial transcriptome profiling by MERFISH reveals subcellular RNA compartmentalization and cell cycle-dependent gene expression. *Proc Natl Acad Sci* [Internet]. 2019 Sep 24;116(39):19490–9. Available from: <https://doi.org/10.1073/pnas.1912459116>
2. Buxbaum AR, Haimovich G, Singer RH. In the right place at the right time: visualizing and understanding mRNA localization. *Nat Rev Mol Cell Biol* 2014 162 [Internet]. 2014 Dec 30 [cited 2024 Jun 14];16(2):95–109. Available from: <https://www.nature.com/articles/nrm3918>
3. Ding J, Adiconis X, Simmons SK, Kowalczyk MS, Hession CC, Marjanovic ND, et al. Systematic comparison of single-cell and single-nucleus RNA-sequencing methods. *Nat Biotechnol* [Internet]. 2020;38(6):737–46. Available from: <https://doi.org/10.1038/s41587-020-0465-8>

Dear Dr. Lin Tang,

We are pleased to resubmit the final version of our manuscript titled “Optimizing Xenium In Situ data utility by quality assessment and best practice analysis workflows” for publication in Nature Methods as an Analysis article. We have addressed the comments raised by reviewers as requested. Only Reviewer 1 requested some minor changes or clarifications. Thus, only his comments are included in this document. Here you can find a point-by-point answer to reviewer’s comments.

Reviewer #1:

Remarks to the Author:

- The code is now improved and provides a documentation for the end-to-end analysis of Xenium data. The workflow is implemented in a notebook.

We are pleased to see that our modifications have satisfied the reviewer’s requests.

- The technology comparison focuses primarily on brain structures, which – as mentioned by the author – makes sense since “ground-truth” is available. This should be clearly stated when discussing the comparison and performance of the various platforms. The results may be very different when applied to other tissue types.

We have modified the text according to reviewer’s suggestions. In the revised version of the manuscript we have clarified that our results regarding tissue domain identification might be tissue-dependent.

- Figure 3F: Use random order for visualization and/or improve image quality.

We have generated a new panel with cells plotted in random order.